# Sub-second and ppm-level optical sensing of hydrogen using templated control of nano-hydride geometry and composition

Hoang Mai Luong [1✉], Minh Thien Pham[1], Tyler Guin[2], Richa Pokharel Madhogaria[3], Manh-Huong Phan [3], George Keefe Larsen [2✉] & Tho Duc Nguyen[1✉]

The use of hydrogen as a clean and renewable alternative to fossil fuels requires a suite of flammability mitigating technologies, particularly robust sensors for hydrogen leak detection and concentration monitoring. To this end, we have developed a class of lightweight optical hydrogen sensors based on a metasurface of Pd nano-patchy particle arrays, which fulfills the increasing requirements of a safe hydrogen fuel sensing system with no risk of sparking. The structure of the optical sensor is readily nano-engineered to yield extraordinarily rapid response to hydrogen gas (<3 s at 1 mbar $H_2$) with a high degree of accuracy (<5%). By incorporating 20% Ag, Au or Co, the sensing performances of the Pd-alloy sensor are significantly enhanced, especially for the $Pd_{80}Co_{20}$ sensor whose optical response time at 1 mbar of $H_2$ is just ~0.85 s, while preserving the excellent accuracy (<2.5%), limit of detection (2.5 ppm), and robustness against aging, temperature, and interfering gases. The superior performance of our sensor places it among the fastest and most sensitive optical hydrogen sensors.

[1] Department of Physics and Astronomy, University of Georgia, Athens, GA, USA. [2] National Security Directorate, Savannah River National Laboratory, Aiken, SC, USA. [3] Department of Physics, University of South Florida, Tampa, FL, USA. ✉email: hoanglm@uga.edu; george.larsen@srnl.doe.gov; ngtho@uga.edu

Hydrogen fuel is a key energy carrier of the future, and it is the most practical alternative to fossil fuel-based chemical storage, with a high theoretical energy density and universality of sourcing[1]. However, significant challenges remain with respect to the safe deployment of hydrogen fuel sources and therefore its widespread adoption[2]. For hydrogen leakage detection and concentration controls, it is essential that hydrogen sensors have good stability, high sensitivity, rapid response time, and most importantly be "spark-free"[3,4]. High performance hydrogen sensors are, however, not only of importance in future hydrogen economy but also the chemical industry[5], food industry[6,7], medical applications[8], nuclear reactors[9], and environment pollution control[10].

Numerous optical hydrogen sensors based on hydride-forming metal plasmonic nanostructure have been explored[11,12]. Pd is the most common hydriding metal for sensor applications due to its rapid response, room temperature reversibility, and relative inertness[13]. However, pure Pd nanoparticles suffer from the coexistence of an α–β mixed phase region, inducing hysteresis and hence non-specific readout and limited reaction rate[12,14,15]. It is possible to minimize this hysteresis and boost the reaction kinetics through the incorporation of alloying metals, such as Co, Au, or Ag[11,12,16]. It has been believed that the enhanced reaction kinetics in the alloying metal hydrides is associated with the reduction of the enthalpy of formation due to the reduced abrupt volume expansion/contraction occurring in smaller mixed phase regions, resulting in a reduction of energy barrier for hydride formation/dissociation, and the improved diffusion rate upon (de)hydrogenation[12,15,17–20]. Synergistically with material design, various sensing nanostructures have been engineered to minimize the volume-to-surface ratio (VSR) of the sensor, a critical condition for achieving fast response time and low hysteresis[12]. These structures include nanostripes[21], nanoholes[22,23], lattices[23], nanobipyramids[24], hemispherical caps[23], nanowires, mesowires[25], and chiral helices[26,27]. Moving beyond metals, other materials such as polymers can be incorporated, as Nugroho et al.[11] recently demonstrated, significantly boosting the sensitivity and response time of a Pd-based nanosensor. Along with downsizing the active material layer, the benchmark response time of 1 s at 1 mbar $H_2$ and 30 °C has been achieved[11], however optical contrast was sacrificed. As a result, sub-second response time and ppm limit of detection (LOD) have not been achieved in a single sensor (Supplementary Table 1).

In this work, we demonstrate a compact optical hydrogen sensing platform with the fastest response reported to date and sub-10-ppm LOD. The sensor is comprised of Pd and Pd-alloy nanopatchy (NP) arrays with a simple optical transmission intensity readout. These hexagonally packed nano-arrays are generated by facile single metal glancing angle deposition (GLAD) on polystyrene (PS) nanosphere monolayers. This fabrication process requires just one deposition step with no post-processing required, simplifying scale-up and reducing costs. The tunable film thickness, patchy diameter and shape (hemispheres or donuts), and therefore VSR can be readily controlled, resulting in tunable and rapid response rate. By alloying with cobalt, the response time of the metasurfaces was reduced below 0.85 s from 1 to 100 mbar of $H_2$ partial pressure, surpassing the strictest requirements for $H_2$ sensing while preserving excellent accuracy (<2.5% full scale) and 2.5-ppm LOD. In a broader perspective, our work illustrates evolution in hydrogen gas sensor technologies through rational topological design and targeted integration of non-traditional materials, such as polymers and active alloying elements. These concepts are universal for promoting strong interactions between gas and materials and may be generally applied to advance sensor and catalyst development, among others.

## Results and discussion

**Pd NP-based optical hydrogen sensors.** The fabrication scheme and nanoarchitecture of an optical hydrogen sensor based on a hexagonal array of Pd hemispherical NPs are depicted in Fig. 1a (see Methods and Supplementary Figs. 1–5). A vapor incident angle of $\theta \geq 50°$ was chosen to ensure that the film would not be deposited directly onto the underlying glass substrate (Supplementary Fig. 2)[23]. The designed structure consists of NP arrays on top of hexagonal closed-packed nanosphere monolayer, which is confirmed by SEM and energy-dispersive spectroscopy (EDS) elemental mapping (see Fig. 1b–d and Supplementary Figs. 6–8). Note that NP samples with a specific deposited thickness will now be referred to as $NP_t^\theta$, where $\theta$ (°) and $t$ (nm) represent the incident angle of the metal vapor and the nominal thickness of the deposited film, respectively.

The morphological transition of Pd $NP_{t_{Pd}}^\theta$ with an increasing $t_{Pd}$ was observed using ultra-high-resolution SEM (SU-9000, Hitachi), as revealed in Fig. 1e. The morphology of $NP_{1.5}^{50}$ contains many sub-10-nm granules, and these granules fully cover the top surface of the polystyrene nanosphere. The size of the granules grows when $t_{Pd} = 2$ nm, and coalescence via a neck (or bridge) connection between neighboring clusters[28] can be noticed at the thickness of $t_{Pd} = 3.5$ nm. A continuous film is formed at a thickness of 5 nm, as these bridge connections are successively grown. Once the continuous film is shaped at a thickness of 5 nm, another distinct coalescent mechanism is observed: the size of the clusters increases as the thickness increases, associated with the reduction of the cluster density (see Supplementary Fig. 8 for a detail analysis). The influences of the film morphology on the thermodynamic response of NP samples will be discussed in the sections below.

The transmission magnitude of the Pd metasurfaces monotonically increases and approaches that of a bare PS nanosphere monolayer with decreasing $t_{Pd}$ (Fig. 2a). Effects from the PS monolayer are observed in the transmission spectra of all films, such as the optical band gap located at wavelength $\lambda = 618$ nm and other transmission maxima/minima at shorter wavelengths due to interference effects[29]. Upon hydrogenation, the optical transmission ($T_{P_{H_2}}$) of all NPs increases with a strong dependence on wavelength as shown in Fig. 2b, where $\Delta T(\%) = T_{P_{H_2}} - T_0$ is the difference of transmission intensity when $P_{H_2} = 1000$ and 0 mbar, respectively. The maximum optical change is much greater than that of the control Pd thin film sample (Supplementary Fig. 14). Note that the optical transmission of a stand-alone PS nanosphere monolayer does not change upon its exposure to hydrogen (Supplementary Fig. 14). The increasing trend of optical transparency upon hydrogenation has been also observed in Pd thin films, although this increase is independent of wavelength in the visible–NIR region[30,31].

Optical hydrogen sorption isotherm of $NP_{t_{Pd}}^{50}$ samples are shown in Fig. 2c, where the change of transmission intensity $\Delta T$ (%) versus $P_{H_2}$ is calculated at the spectrum maxima to achieve the highest signal-to-noise ratio (SNR). We note that the plateau pressure could be measured at any wavelength within the visible region. Several important dependencies can be observed from the optical isotherms. Primarily, $\Delta T$ is generally reduced as $t_{Pd}$ decreases due to lower overall light-patch-interaction signal. The mixed region hysteresis is reduced, and the slope of the plateau pressure increases with decreasing $t_{Pd}$. The size dependences of the plateau slope and hysteresis have been previously observed in Pd nanoparticles with a narrow size distribution, but we cannot discount the effect of size distribution on the slope for $NP_{t_{Pd}}^{50}$ samples (Supplementary Figs. 8–9)[32].

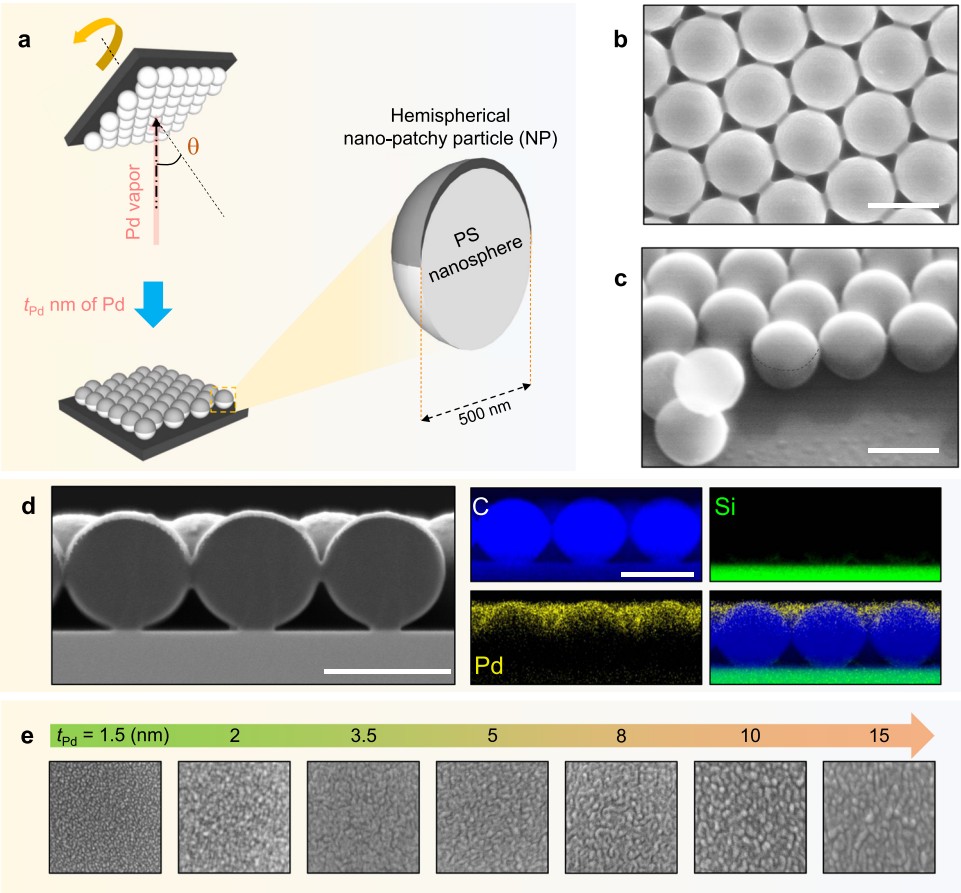

**Fig. 1 Fabrication scheme and morphology characterization. a** Schematic of the fabrication process. **b** Top-view and **c** side-view scanning electron microscope (SEM) images of Pd $NP_{15}^{50}$. Scale bars correspond to 500 nm. **d** Cross-sectional SEM image and EDS elemental maps of $NP_5^{50}$ showing a metal cap formed on the top surface of a PS nanosphere. Scale bars correspond to 500 nm. **e** Magnified views (150 nm × 150 nm) of ultra-high resolution SEM micrographs (in Supplementary Fig. 8) showing morphology differences of Pd NP with different deposited thickness ($t_{Pd}$) of 1.5 to 15 nm.

Figure 2d displays the plateau pressure of absorption isotherm, desorption isotherm, and hysteresis ($P_{Abs}$, $P_{Des}$, and $\ln(P_{Abs}/P_{Des})$, respectively) versus $t_{Pd}$ (the extraction method is described in Supplementary Fig. 16). We observe a critical thickness ($t_{Pd} \approx 5$ nm) above which the nanoparticles behave qualitatively differently than for thicknesses below. It is worthwhile to note that this critical thickness corresponds well with the expected transition from a separated island-like morphology to a percolating film morphology. Above this critical thickness, $P_{Abs}$ and $P_{Des}$ are relatively independent of $t_{Pd}$ and $\ln(P_{Abs}/P_{Des}) = 0.62$, which is the value predicted for bulk Pd undergoing a coherent phase transition[33]. Below the critical thickness, both $P_{Abs}$ and $P_{Des}$ sharply increase, while $\ln(P_{Abs}/P_{Des})$ sharply decreases. This change in $P_{Abs}$ and $P_{Des}$ is due to the increasing effects of higher energy subsurface adsorption sites relative to bulk sites[32,34]. The significant decrease in hysteresis for shrinking particle size at a constant temperature of Pd material, $T$, has been widely observed and is understood as a decrease in critical temperature, $T_c$, with decreasing the particle size (Supplementary Fig. 11)[35]. In general, when $T$ is close to $T_c$, the hysteresis behavior is strongly suppressed. See Supplementary Figs. 11–13 for more in-depth analyses of $P_{Abs}$, $P_{Des}$, and $\ln(P_{Abs}/P_{Des})$ as functions of $t_{Pd}$.

**Performance of Pd metasurfaces.** The hysteresis of an optical sensor, and therefore its insensitivity to measurement history, can be calculated in terms of "sensor accuracy", as proposed by Wadell et al. (Supplementary Equation (20))[36]. Figure 2e

summarizes the sensor accuracy of $NP_{t_{Pd}}^{50}$ with varying $t_{Pd}$, from $10^1$ to $10^6$ µbar $P_{H_2}$. The $P_{Hys}$ ($= P_{Abs} - P_{Des}$) reported here (~7.5 mbar in $NP_{15}^{50}$, ~4 mbar in $NP_2^{50}$) is significantly smaller than $P_{Hys}$ reported for any other optical based nanostructure/thin film sensing platforms made from pure Pd (typically $P_{Hys} > 20$ mbar)[11,36]. Therefore, these NP-based sensors offer very high sensor accuracy (<10 % at $t_{Pd} \leq 15$ nm and <5 % at $t_{Pd} \leq 5$ nm), which is comparable to that of Pd-alloy-based hydrogen sensors[36,37].

The response time of the NP sensor is improved with decreasing $t_{Pd}$. The response times ($t_{90}$, the time required to reach 90% of the final equilibrium response) of $NP_{t_{Pd}}^{50}$ sensors with varying $t_{Pd}$ were measured when varying pressure pulses from 100 to 1 mbar $H_2$. The $t_{90}$ generally decreases with increasing $P_{H_2}$, though there is a peak at $P_{Abs}$ (Fig. 2f). The peak in the $t_{90}$ versus $P_{H_2}$ relation for $P_{Abs}$ has been reported in some Pd-based hydrogen sensor[23,38,39]; however, the origin of this observation has not been described. We hypothesize that the peak is associated to the abrupt change in hydride volume during the $\alpha - \beta$ phase transition. Such abrupt change or sudden entropy change occurs only if the hydride system thermodynamically overcomes the large strain-induced energy barrier upon hydrogenation, which requires longer time to reach the equilibrium. The strain-induced energy barrier is relatively strong in pure Pd based sensors, that results in the large $t_{90}$ peak in this pressure regime. It is clear in Fig. 2f that such barrier is larger with a larger grain size and film thickness[12], and interestingly, the barrier

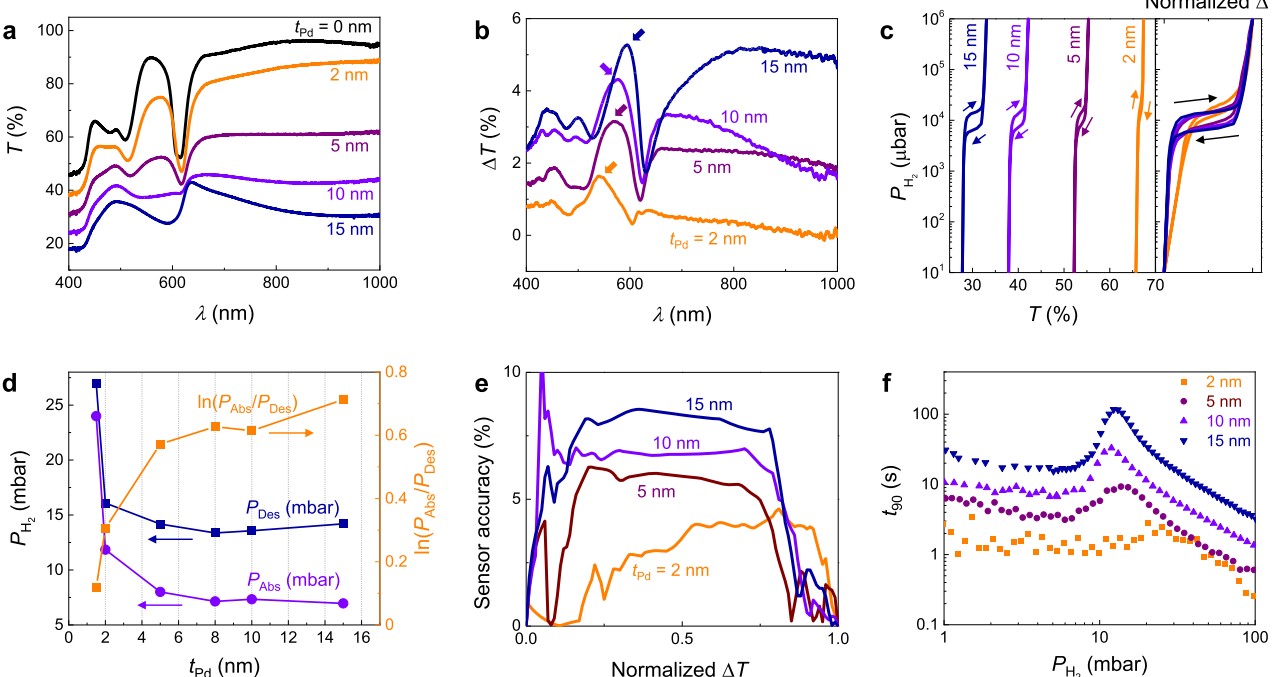

**Fig. 2 Pd NP$_{t_{Pd}}^{50}$ sensors with different $t_{Pd}$. a** Experimental optical transmission spectra $T(\lambda)$ (at $P_{H_2}$ = 0 mbar) and **b** optical transmission changes $\Delta T(\lambda)$ = $T_{1000mbar}$ - $T_{0mbar}$ of NP with different $t_{Pd}$. Colored arrows indicate $\Delta T(\lambda)$ maxima. **c** Optical hydrogen sorption isotherm of NP with different $t_{Pd}$, extracted at $\Delta T(\lambda)$ maxima. The panel to the right shows corresponding normalized $\Delta T(\lambda)$ hydrogen sorption isotherm of NP with different $t_{Pd}$. Arrows denote the sorption direction. **d** Extracted plateau pressures for hydrogen absorption ($P_{Abs}$), desorption ($P_{Des}$), and $\ln(P_{Abs}/P_{Des})$ for different $t_{Pd}$. **e** Sensor accuracy at specific normalized $\Delta T$ readout over hydrogen pressure range of $10^1$ μbar to $10^6$ μbar. **f** Response time of NP sensors with pulse of hydrogen pressure from 1 mbar to 100 mbar.

becomes much less significant as the structure transforms from film-like to nanoparticle island-like for $t_{Pd} \leq 2$ nm. The $t_{90}$ of NP$_2^{50}$ is <3 s over the 1–100 mbar range, which is nearly two orders of magnitude smaller than that of NP$_{15}^{50}$ (the longest $t_{90}$ is ~120 s). Crucially, the $t_{90}$ of NP$_2^{50}$ and NP$_5^{50}$ samples at 40 mbar (the lower flammability limit of hydrogen) is ~1 s, which is shorter than that of previously reported pure Pd-based optical sensors (~7.5 s[11], or typically >20 s)[38], and comparable to that of Pd-alloy based sensors[36,37]. In general, the response time as a function of $t_{Pd}$ follows a power-law scaling with respect to the VSR (VSR$^z$), with an exponent of $z = 2.3$, which agrees well with a diffusion limited metal hydride system (Supplementary Fig. 13)[40,41].

The flexibility of the GLAD technique allows the realization of different nanopatterns of patchy array sample by adjusting the polar and azimuthal angle of deposition onto the packed nanosphere monolayers (Supplementary Figs. 2–3)[42]. Here, we utilize $\theta = 70°$, $80°$, and $85°$ with constant azimuthal rotation (fixed $t_{Pd} = 15$ nm) to engineer the morphology and VSR of NPs. Figure 3a displays SEM micrographs of NP$_{15}^{50}$, NP$_{15}^{70}$, NP$_{15}^{80}$, and NP$_{15}^{85}$ samples, which confirm the predicted surface morphologies. Figure 3b displays the optical hydrogen sorption isotherm of these samples at maximum $\Delta T(\lambda)$. The amount of material deposited decreases as $\theta$ increases, and correspondingly, the samples are generally more transparent and $\Delta T$ changes less upon hydrogenation (Fig. 3b). Apart from the transmission magnitude change, we observe a narrowing of hysteresis in the mixed phase region and an increasing slope of plateau as $\theta$ increases. The phase transition behavior of NP is further shown in Supplementary Fig. 18.

As mentioned, a key advantage of this nano-fabrication method is the straightforward engineering of the VSR, which can decrease sensor response times[11]. By increasing $\theta$, one can

efficiently reduce VSR since the patchy pattern gradually transforms from hemisphere-like to donut-like structures. A strong $\theta$-dependent hydriding kinetics of NP$_{15}^{\theta}$ can be observed in Fig. 3c. For example, the $t_{90}$ of a NP$_{15}^{85}$ is <10 s over pressures 1–100 mbar, which is nearly an order of magnitude smaller than these of a NP$_{15}^{50}$. More significantly, we observe a general linear relation between VSR and $t_{90}$ at different $P_{H_2}$ of NP sample, regardless of vapor incident angle $\theta$ and deposited thickness $t_{Pd}$ (Fig. 3d). Note that the estimation of VSR is based on simulated morphologies[43], (Supplementary Figs. 1–5) and there may be some deviations from the achieved parameters due to non-idealities, surface roughness, and island morphologies. However, the general trend of creating faster absorption kinetics was observed when the VSR and/or $t_{Pd}$ was reduced. Additionally, we highlight that the VSR may be further optimized to achieve even shorter response time by other nanoparticle forms, such as nano-fan[42], which can be created by modifying depositions onto packed nanosphere monolayers with different polar and azimuthal angles.

**Pd-composite NP optical hydrogen sensors.** To further optimize the accuracy and response time of the nano-sensor, we utilized Pd-based composites with alloying elements (Ag, Au, or Co) to modify the hydriding properties of Pd. For example, PdAg and PdAu alloys display lower plateau pressures than pure Pd[11,23,36,44], while PdCo alloys display much higher plateau pressures[45,46]. Pd$_{80}$Ag$_{20}$, Pd$_{80}$Au$_{20}$, and Pd$_{80}$Co$_{20}$ composite NP$_5^{50}$ and NP$_{15}^{50}$ samples were fabricated by employing the electron beam co-evaporating method. It is worth noting that the Pd$_{80}$Au$_{20}$ sensors provide a useful direct comparison to the major works on PdAu hydride systems[16,36,47–49]. Utilizing EDS

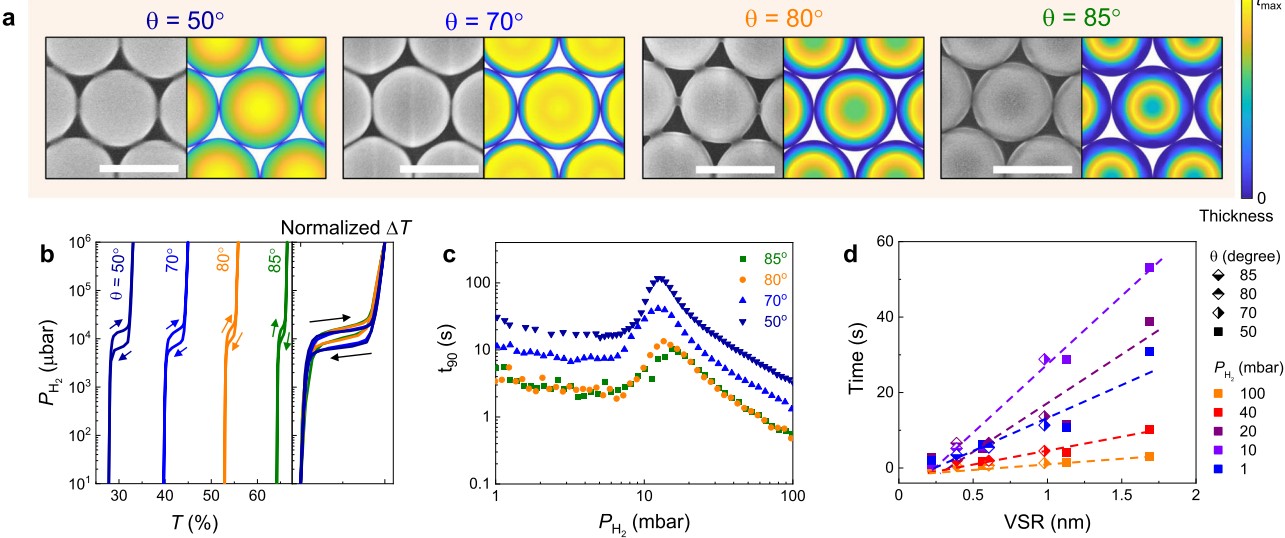

**Fig. 3 Pd NP$_{15}^{\theta}$ sensors with different $\theta$. a** Top-view SEM images (left) and corresponding simulated morphologies (right) of pure Pd NP samples, fabricated with vapor incident angle of $\theta = 50°$, $70°$, $80°$, and $85°$, respectively. Scale bars correspond to 500 nm, and the color bar shows the Pd thickness distribution in the simulated hemisphere cap. **b** Optical hydrogen sorption isotherm of NP with different $\theta$, extracted at $\Delta T(\lambda)$ maxima. Arrows denote the sorption direction. The panel to the right shows corresponding normalized $\Delta T(\lambda)$ hydrogen sorption isotherm of NP with different $\theta$. **c** Response time of NP sensors with pulse of hydrogen pressure from 1 mbar to 100 mbar. **d** Response time of NP samples as a function of volume-to-surface ratio of NP sensors, at different hydrogen pressure. Note that the optical isotherm data of NP$_{15}^{50}$ sample presented in Figs. 2c and 3b are identical, and response time data of NP$_{15}^{50}$ sample presented in Figs. 2f and 3c are identical.

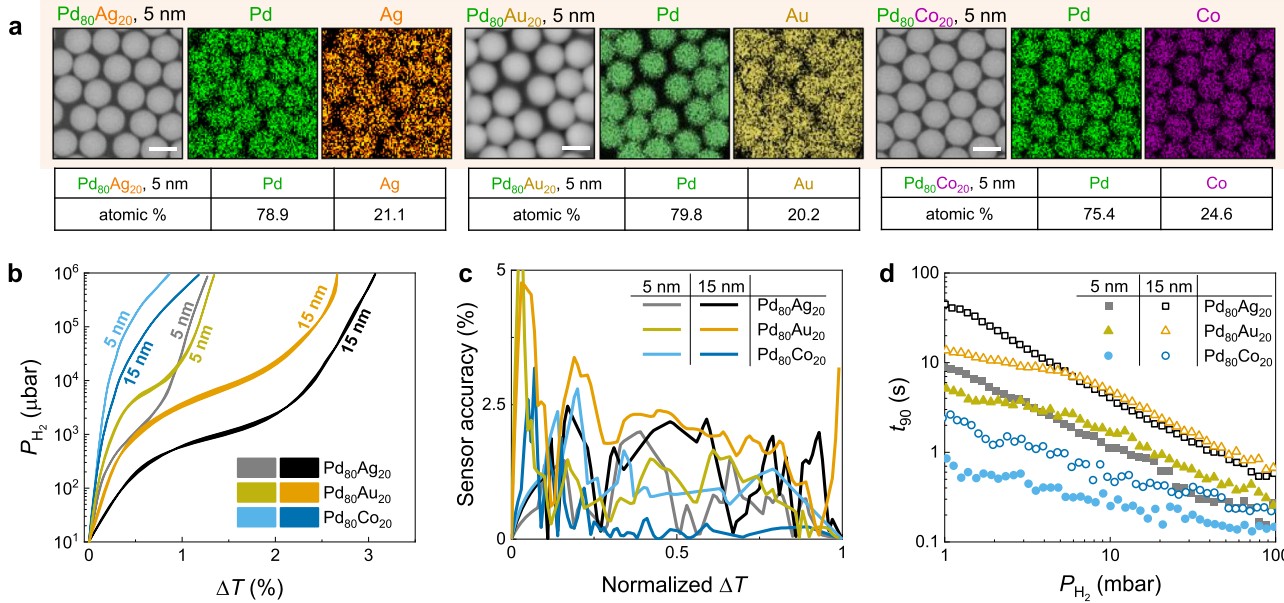

**Fig. 4 Composite NP sensors. a** SEM and EDS images of Pd$_{80}$Ag$_{20}$, Pd$_{80}$Au$_{20}$, and Pd$_{80}$Co$_{20}$ ($t = 5$ nm) composite NP samples. Scale bars correspond to 500 nm. Tables show the elemental atomic composition (at. %) of the Pd$_{80}$Ag$_{20}$ and Pd$_{80}$Co$_{20}$ NPs, which are consistent with the desired compositions. **b** Optical hydrogen sorption isotherm of composite NP, extracted at $\Delta T(\lambda)$ maxima. **c** Sensor accuracy of composite NP sensors at specific normalized $\Delta T$ readout over hydrogen pressure range of $10^1$ μbar to $10^6$ μbar. **d** Response time of NP sensors with pulse of hydrogen pressure from 1 mbar to 100 mbar.

elemental mapping, we confirm the composition and uniform element distribution of Pd and Ag/Au/Co in the composite NP samples (Fig. 4a).

The hydrogen sorption characteristics of the Pd$_{80}$Ag$_{20}$ alloy NP system are examined as previously (Supplementary Fig. 19). The plateau of Pd$_{80}$Ag$_{20}$ NP is shifted downward to lower pressure ($P_{Abs} = 1$ mbar) and shows an increasing slope in comparison to that of the Pd NP (Supplementary Fig. 19c). The hysteresis is very narrow ($P_{Hys}$ are <0.1 mbar and <0.05 mbar in Pd$_{80}$Ag$_{20}$ NP$_{15}^{50}$

and NP$_{5}^{50}$, respectively), and the optical isotherm could be considered "hysteresis-free" (Fig. 4b). These behaviors of Pd$_{80}$Ag$_{20}$ NP system are in agreement with bulk PdAg alloy[44], and are similar to behaviors of PdAu alloy nanoparticles and thin films with higher Au content[36,37,50]. We note that the $P_{Abs}$ of PdAg composite NP ($P_{Abs} = 1$ mbar) is significantly smaller than those of the previously employed systems (typically $P_{Abs} \geq 10$ mbar)[36,37,50]. In addition, the absorption kinetics are noticeably accelerated as the $t_{90}$ of Pd$_{80}$Ag$_{20}$ sensors are significantly faster

than that of Pd sensors with the same deposited thickness (Fig. 4d). The origin of this acceleration could be the reduction of the apparent activation energy for hydrogen sorption due to combination with Ag[51], change in morphology to well-separated islands (Supplementary Fig. S10), and due to a decrease in abrupt volume expansion occurring in a smaller mixed phase region, as observed in the narrow hysteresis[52], both leading to a significant increment of hydrogen permeability in $Pd_{80}Ag_{20}$[53]. The pressure-dependent response time of $Pd_{80}Ag_{20}$ sensors follows the power-law[11] as confirmed by an near-linear dependence between $\log(t_{90})$ versus $\log(P_{H_2})$.

In comparison to PdAg system, PdAu samples with the same thickness exhibits a similar spectra shape and transmission magnitude (Supplementary Fig. 19a). Upon the hydrogenation, $Pd_{80}Au_{20}$ $NP_5^{50}$ and $NP_{15}^{50}$ samples behave similar to $Pd_{80}Ag_{20}$ $NP_5^{50}$ and $NP_{15}^{50}$, with comparable optical transmission changes ($\Delta T(\lambda)$) and similar spectra shape (Supplementary Fig. 19b). However, the plateau pressures found in $\Delta T(\lambda)$ optical isotherms of PdAu samples are at higher pressure regimes ($P_{Abs} = 7$ mbar) than these of PdAg systems ($P_{Abs} = 1$ mbar) (Fig. 4b). We note that the $P_{Abs}$ we found in PdAg NPs are slightly smaller than that of PdAu nanoparticles with a similar composition ($P_{Abs} \geq 10$ mbar)[36,37,50]. In addition, a noticeable hysteresis can be seen in $\Delta T(\lambda)$ isotherms of $Pd_{80}Au_{20}$ $NP_{15}^{50}$ ($P_{Hys} = 1.1$ mbar) and $Pd_{80}Au_{20}$ $NP_5^{50}$ ($P_{Hys} = 0.7$ mbar), which are wider than these of PdAg samples ($P_{Hys}$ are $<0.1$ mbar). However, the sensor accuracies calculated over the pressure range of $10^1$ to $10^6$ μbar are still below 5% level (Fig. 4c). At a given pressure $P_{H_2} > P_{Abs} = 7$ mbar, response time ($t_{90}$) of a PdAu sensor is slightly higher than that of a PdAg sensor with the same thickness, which follows the power-law with a similar slope (Fig. 4d). Interestingly, we observe a kink at $P_{Abs} = 7$ mbar, below which $t_{90}$ of PdAu still follows the power law, however, with a noticeable smaller slope. The $t_{90}$ of $Pd_{80}Au_{20}$ $NP_{15}^{50}$ and $NP_5^{50}$ samples at 1 mbar are 13 s and 5 s, respectively, which are much shorter than these of PdAg sensors (45 s and 9 s for $NP_{15}^{50}$ and $NP_5^{50}$ samples, respectively). Since the kink is observed in the middle of the plateau, which is in a similar position as that in the pure Pd sensor case (Fig. 2f), we hypothesize that the strain-induced energy barrier in the $Pd_{80}Au_{20}$ is still significant at $P_{Abs} = 7$ mbar and is the cause of this observation. We note that the differences in $t_{90}$ between the PdAu and PdAg samples with the same $t$ and $P_{H_2}$ might come from several factors, such as differences in morphologies (see Supplementary Figs. 9–10), and hydrogen permeability[54].

Several advantages are achieved through the utilization of a PdAg and PdAu composite instead of pure Pd in a representative $NP_{15}^{50}$ sensor, which are listed as (1)–(4). (1) The "hysteresis-free" optical isotherm removes undesirable uncertainty, generating an excellent sensor accuracy of $<5\%$ over the pressure range of $10^1$ to $10^6$ μbar (Fig. 4c). (2) The slope of absorption isotherm of PdAg and PdAu samples in the α-phase ($P_{H_2} < 0.5$ mbar) shows a 3- and 1.8-times enhancement in sensitivity as compared to pure Pd sensors, respectively (Supplementary Fig. 19c), which benefits trace level detection. (3) Particularly for PdAg system, the plateau shifts to very low pressures ($P_{Abs} = 1$ mbar), which enhances the sensitivity by an order of magnitude throughout the pressure range of 0.5 mbar to 10 mbar (~0.05–1 vol%). This is essential for early leak detection (4 vol% is the lower flammability limit of hydrogen). (4) The absorption kinetics are accelerated (Fig. 4d), which yields an order of magnitude $t_{90}$ faster than that of pure Pd at the plateau pressure.

While the advantages (1)-(3) achieved by $Pd_{80}Ag_{20}$ and $Pd_{80}Au_{20}$ $NP_{15}^{50}$ sensors are comparable to or surpasses the performances of up-to-date optical composite-based nano-sensors[11,16,36,37], the

accelerated response time (~45 s and ~13 s at $P_{H_2} = 1$ mbar, respectively) in (4) is still too slow to fulfill the most rigorous requirement of a hydrogen sensor for automotive applications[3]. Engineering the VSR by downsizing NP significantly reduces this gap, as observed in the $Pd_{80}Ag_{20}$ and $Pd_{80}Au_{20}$ $NP_5^{50}$ sample. While the shape of optical sorption isotherm is relatively unchanged and the sensor accuracy of $<5\%$ is preserved upon thickness reduction from 15 nm to 5 nm (Fig. 4b,c), the $t_{90}$ of $Pd_{80}Ag_{20}$ and $Pd_{80}Au_{20}$ $NP_5^{50}$ is reduced by a factor of ~5 and ~2.6 over a pressure range 1–100 mbar and is only ~9 s and ~5 s at $P_{H_2} = 1$ mbar, respectively. The slopes of the linear fits in ($\log(t_{90})$ versus $\log(P_{H_2})$) plot of $NP_{15}^{50}$ and $NP_5^{50}$ samples are equivalent for both PdAg and PdAu systems (Fig. 4d), which implies that these two samples have similar sorption behaviors and that the improvement of the response time is primarily derived from the reduction of VSR[11]. Hence, in order to achieve the $t_{90} \leq 1$ s at $P_{H_2} = 1$ mbar, the deposited thickness of $Pd_{80}Ag_{20}$ and $Pd_{80}Au_{20}$ should be $<1$ nm (see Supplementary Figs. 1–5 and Supplementary Fig. 13). A minimum deposited thickness of $t = 1.5$ nm is required for overcoming the bead's roughness and obtaining a detectable optical signal contrast in the pure Pd $NP_t^{50}$ at $P_{H_2} = 1$ mbar. This means that it would be impossible to attain a $Pd_{80}Ag_{20}$ $NP_{15}^{50}$ sensor with $t_{90} \leq 1$ s at $P_{H_2} = 1$ mbar simply by reducing $t$.

In order to reduce the response time further, the sorption behavior of the NP sensors was modified through the incorporation of Co ($Pd_{80}Co_{20}$), analogous to $Pd_{80}Ag_{20}$ and $Pd_{80}Au_{20}$. A PdCo alloy improves the kinetics of hydrogenation over a PdAg and PdAu alloy by (1) remaining in the α-phase over the pressure region of interest, which removes the kinetic steps of the α- to β-phase transition and subsequent hydrogen atomic diffusion through the β-phase[55]; and (2) cobalt offers a greater metal-hydrogen bond strength than silver, which could facilitate dissociative chemisorption of hydrogen[56]. It is also worth noting that the comparison between the PdAu and PdCo NP sensors allows isolation of the alloying element effect since their morphologies are very similar (Supplementary Fig. S10). In the $Pd_{80}Co_{20}$ sensors, the overall shape of $\Delta T(\lambda)$ is similar to those of the Pd, $Pd_{80}Ag_{20}$, and $Pd_{80}Au_{20}$ sensors. However, the magnitude of $\Delta T$ is significantly smaller as PdCo alloy has lower H-solubility due to lattice contraction (Supplementary Fig. 19)[45].

Figure 4b shows the $\Delta T - P_{H_2}$ isotherm of $Pd_{80}Co_{20}$ $NP_5^{50}$ extracted at the maximum $\Delta T(\lambda)$. The plateau pressure is shifted significantly upward, and consequently, no hysteresis in the sorption isotherm is observed in the measured pressure range, which is consistent with the previous observation for bulk PdCo alloys and similar to the behaviors of the PdCu alloy nano-sensors[37,45,46]. The "hysteresis-free" characteristic is reflected in the very high sensor accuracy ($<2.5\%$) (Fig. 4c). Additionally, remarkably accelerated absorption in the PdCo composite is observed (Fig. 4d). Two notable results are emphasized: (1) the $t_{90}$ at $P_{H_2} = 1$ mbar is 0.85 s, which satisfies the most stringent requirement of a hydrogen sensor ($<1$ s over pressures 1–100 mbar), matching the fastest optical hydrogen nano-sensor under similar conditions[11], and (2) the $t_{90}$ at $P_{H_2} = 40$ mbar (~4 vol %) is ~0.15 s (the resolution limit of our system at a 12.5 Hz sampling rate is at 0.15 s), which is nearly twice as fast as those of previously reported optical hydrogen nano-sensors under similar conditions[11,16,36,37]. In addition, we highlight that (i) coating of a polymer thin film and (ii) optimizing the VSR may further improve this ultra-fast response time[11].

One shortcoming of downsizing the active material layer is that the optical contrast upon (de)hydrogenation decreases significantly, which makes it challenging to achieve ultra-fast response

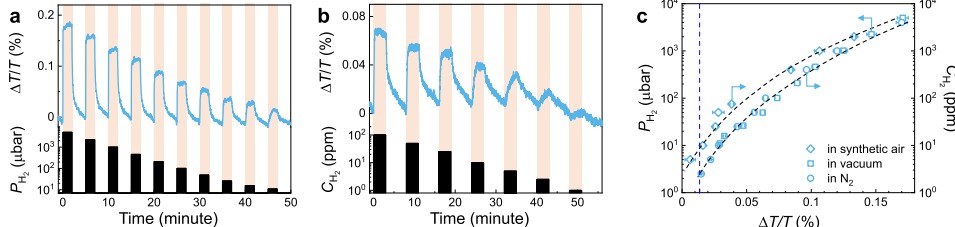

**Fig. 5 Sensing performances of composite PdCo NP sensors. a** $\Delta T/T$ response of $Pd_{80}Co_{20}$ $NP_5^{50}$ sensors to stepwise decreasing hydrogen pressure in the 5000–11 μbar range, measured at 1.25 Hz sampling frequency in a vacuum chamber. **b** $\Delta T/T$ response of $Pd_{80}Co_{20}$ $NP_5^{50}$ sensors with different hydrogen concentrations ($C_{H_2}$) of 100–1 ppm, measured in flowing nitrogen (400 ml/min). Shaded areas denote the periods where the sensor is exposed to hydrogen. **c** Measured $\Delta T/T$ response as a function of hydrogen pressure/concentration derived from **a**–**b** and Supplementary Figs. 20–21. The blue dashed line denotes the defined LOD at $3\sigma = 0.013$ % (see Supplementary Fig. 26).

time and ultra-low LOD in a single nano-sensor. The unique design of the NP sensor allows for a very high surface coverage (>90%, Supplementary Fig. 17) and results in sizable optical contrast even at very low concentration of hydrogen. In Fig. 5a, the detection capability of the PdCo $NP_5^{50}$ hydrogen sensor is demonstrated in step-wise pressure pulses of pure $H_2$ from 5000 to 11 μbar at a 1.25 Hz sampling rate (11 μbar is the lower pressure limit of our measurement setup). Clearly, PdCo $NP_5^{50}$ can resolve the lowest pulse and potentially provide detection at even lower hydrogen pressures. The PdCo $NP_5^{50}$ sensor is further tested in flow-mode with different hydrogen concentrations ($C_{H_2}$) from 100 to 1 ppm in nitrogen (Fig. 5b), and 3-cycles of each $C_{H_2}$ from 40000 to 10 ppm in nitrogen (Supplementary Fig. 20). The results show reproducible responses that are distinct from the background signal at $C_{H_2}$ as low as 10 ppm, with the possible experimental LOD as low as 2.5 ppm (Supplementary Figs. 26–27). We note that results for similar tests with PdAg $NP_{15}^{50}$ and $NP_5^{50}$ sensors can be seen in Supplementary Fig. 22. Further sensing test with synthetic air as a carrier gas shows that the PdCo $NP_5^{50}$ sensor exhibits a smaller response at a given $C_{H_2}$, in comparison to the responses of the sensor in vacuum/inert carrier gas (Supplementary Fig. 21). This is due to the competing processes of the hydrogen oxidation reaction and the adsorption sites are blocked by O[37,38]. The $\Delta T(\lambda)/T$ amplitude of the sensors is extracted and summarized in Fig. 5c, which demonstrates approximately equivalent optical responses in vacuum-mode sensing and flow-mode sensing with nitrogen carrier gas, and smaller responses in flow-mode sensing with synthetic air carrier gas. By defining LOD = $3\sigma$, where $\sigma$ is the noise of the acquired signal (which is 0.0042%, see Supplementary Fig. 26), we achieved the LOD of 2.5 ppm in nitrogen and 10 ppm in air. These experimental performances (LOD of 2.5 ppm and $t_{90} = 0.85$ s at 1 mbar, in a single sensor) places our PdCo $NP_5^{50}$ system among the fastest and most sensitive optical hydrogen sensors (see Supplementary Table 1)[11,39].

In order to assess the practical implementation of these hydrogen sensors, the influence of temperature, moisture, and interfering gases (e.g., CO, $CO_2$, $CH_4$) on the sensor performances were investigated. For the PdCo $NP_5^{50}$ sensor, we found that neither 5% $CO_2$ nor 5% $CH_4$ in the 2% $H_2$ feed gas affects to the sensor signal (Fig. 6a, b). In addition, PdCo $NP_5^{50}$ sensor shows a good tolerance toward the temperature, as the signal amplitude stays at ~80% of the reference signal, up to 315 K (Fig. 6c, e). However, a moderate sensor poisoning occurs upon the exposure to the 0.2% CO in the feed gas, as the signal amplitude decreases to ~60% of the reference signal (Fig. 6a, b). We note that the sensor signal can be fully recovered by multiple

(de)hydrogenation cycles (Supplementary Fig. 29). In addition, we observed a ~30% drop in signal amplitude when the synthetic gas carrier air has a relative humidity (RH) of 40% (Fig. 6d, f). This desensitization is understandable, as the water molecules are in contact and dissociatively adsorbed by the Pd-surface, which induces the consumption of $H_2$ and forms the gas phase water[57]. In term of sensor stability upon cycling, the sensor signal is reproducible upon several hundred of (de)hydrogenation cycles without any sign of degradation (Supplementary Fig. 27a, b). However, the sensor signal slowly changes upon long-term storing in the ambient conditions, as the sensor does not completely immune from trace gases present in air, such as CO and moisture. We observed a reduction of sensor signal, as well as decay performances in response time (still, the $t_{90}$ and LOD in nitrogen background of PdCo $NP_5^{50}$ are <2.5 s (at 1 mbar) and <10 ppm, respectively, over a >10-month period and >400 (de) hydrogenation cycles) (Supplementary Fig. 27c–e). Nevertheless, a protective layer, such as a polymer-coating, would protect the sensor against humidity, CO, and other toxic gases and ensure long-term reliability.

Inspired by previous works[11,58,59], our preliminary results demonstrate that a simple polymer coating layer of polymethyl methacrylate (PMMA) achieved by spin-coating can provide excellent protection for PdCo $NP_5^{50}$ sensor against CO and humidity. The standard sensing characterizations of this PdCo $NP_5^{50}$/PMMA sensor can be found in Supplementary Fig. 30. In both deactivation tests with poisonous gases ($CH_4$, $CO_2$, CO, in Fig. 7a, b) and humidified carrier gases (RH = 40%, in Fig. 7c, d), the absolute response of the sensor remains within the ± 20% deviation limit, which satisfy the standard requirement for hydrogen sensors[60]. We note that in this case, the PMMA polymer layer protects the Pd-adsorption domain from direct contact with water molecules, which reduces the $H_2O$-induced $H_2$ consumption and prevents the desensitization[57]. The water vapor condensing on the polymer coating does have an effect by impeding $H_2$ reaching to adsorption domain, and slightly increasing the response/releasing times (Fig. 7c)[61,62]. Notably, the polymer coating layer significantly mitigates the degradation of sensing performance when this sensor operates in a practical condition (Supplementary Fig. 28). This is an interesting improvement for this hydrogen sensing platform that warrants further study.

In summary, we have demonstrated a method to produce a class of rapid-response, highly sensitive, and accurate optical Pd-alloy hydrogen sensors through GLAD on PS-nanospheres. It is facile to tune these metasurface sensors through simple alloying, angle of deposition, or film thickness, which dictates the qualitative nature, quantitative metrics, and hysteresis of the response. By incorporating 20% Co, the sensor response time ($t_{90}$)

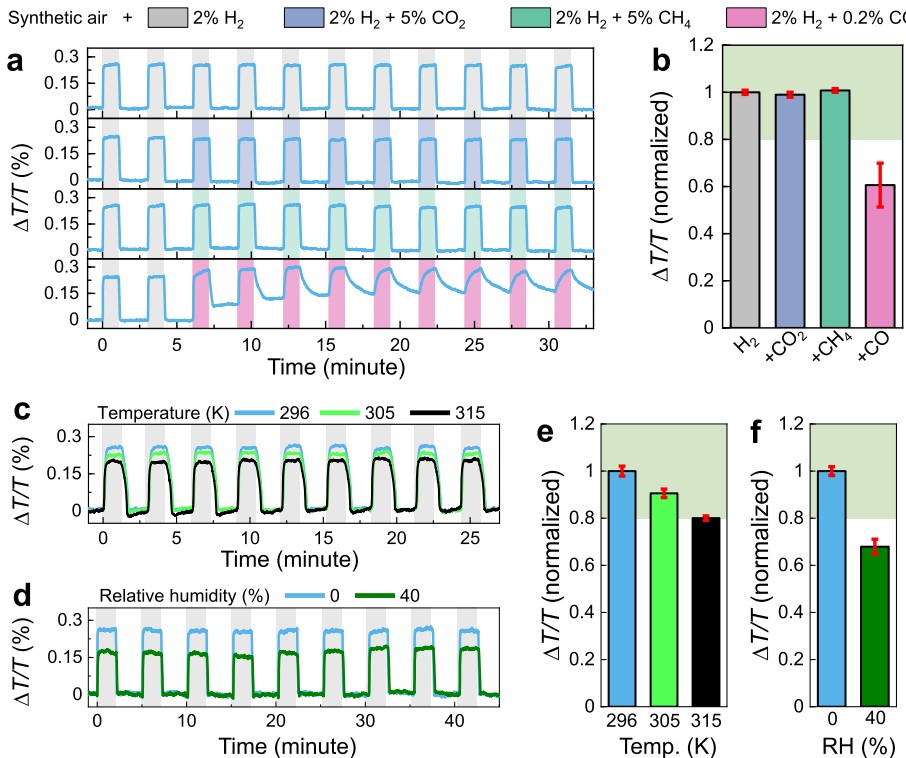

**Fig. 6 Sensing performances of composite PdCo NP sensors. a** Time-resolved $\Delta T/T$ response of $Pd_{80}Co_{20}$ $NP_5^{50}$ to 2 pulses of 2% $H_2$ followed by 9 pulses of 2% $H_2$ + 5% $CH_4$, 2% $H_2$ + 5% $CO_2$, and 2% $H_2$ + 0.2% CO; and **b** normalized sensor signal to the one obtained with 2% $H_2$ in synthetic gas flow. The error bars denote the standard deviation from 9 cycles. **c** Time-resolved $\Delta T/T$ response of $Pd_{80}Co_{20}$ $NP_5^{50}$ to 9 pulses of 2% $H_2$ with different temperature and **e** normalized sensor signal one obtained with 2% $H_2$ in synthetic gas flow at 296 K. **d** Time-resolved $\Delta T/T$ response of $Pd_{80}Co_{20}$ $NP_5^{50}$ to 9 pulses of 2% $H_2$ with different relative humidity (RH) and **f** normalized sensor signal one obtained with 2% $H_2$ in dry condition. All measurements were performed using synthetic gas as carrier gas. The green shaded areas in **b**, **e**, and **f** indicate the ±20% deviation limit from the normalized $\Delta T/T$ response with 2% $H_2$.

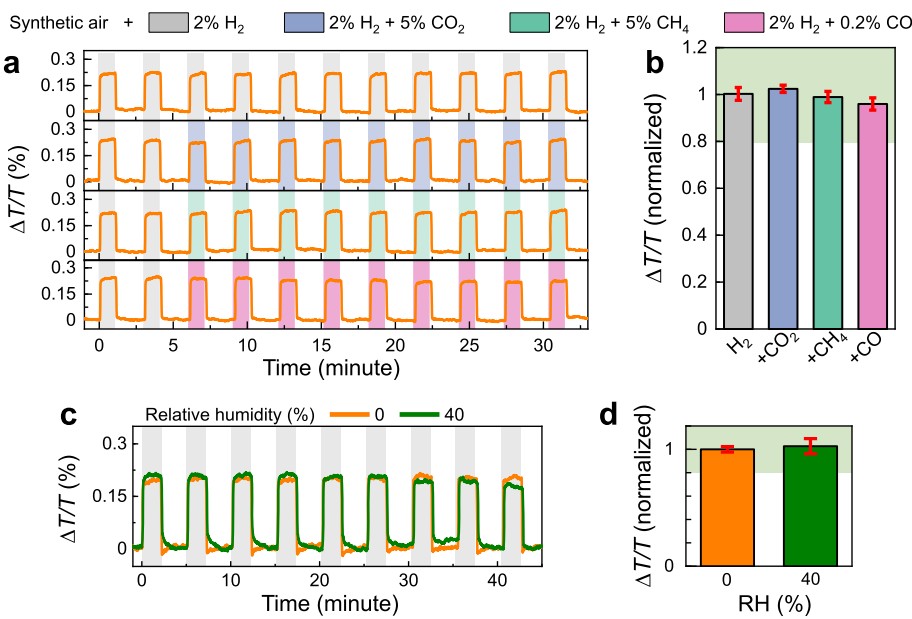

**Fig. 7 Sensing performances of composite PdCo NP/PMMA sensors. a** Time-resolved $\Delta T/T$ response of $Pd_{80}Co_{20}$ $NP_5^{50}$/PMMA to 2 pulses of 2% $H_2$ followed by 9 pulses of 2% $H_2$ + 5% $CH_4$, 2% $H_2$ + 5% $CO_2$, and 2% $H_2$ + 0.2% CO; and **b** normalized sensor signal to the one obtained with 2% $H_2$ in synthetic gas flow. The error bars denote the standard deviation from 9 cycles. **c** Time-resolved $\Delta T/T$ response of $Pd_{80}Co_{20}$ $NP_5^{50}$/PMMA to 9 pulses of 2% $H_2$ with different RH and **d** normalized sensor signal one obtained with 2% $H_2$ in dry condition. All measurements were performed using synthetic gas as carrier gas. The green shaded areas in **b** and **d** indicate the ±20% deviation limit from the normalized $\Delta T/T$ response with 2% $H_2$.

at 1 mbar is less than 0.85 s, which is the fastest response ever reported at the critical $H_2$ concentration required for leak detection. Additionally, these sensors readily detect concentration as a low as 2.5 ppm in nitrogen and 10 ppm in air, maintain high accuracy due to the hysteresis-free operation, and exhibit robustness against aging, temperature, moisture, and interfering gases. These sensors demonstrate a viable path forward to spark-proof optical sensors for hydrogen detection applications.

## Methods

**Materials**. Polystyrene (PS) nanospheres (Polysciences Inc., $D = 500 \pm 10$ nm) and ethanol (Sigma-Aldrich, 98%) were used to create the nanosphere monolayers. Palladium (99.95%), silver (99.99%), and cobalt (99.95%) from Kurt. J Lesker Company were utilized for electron beam depositions. Deionized water (18 MΩ cm) was used for all experiments.

**Sample fabrication**. Hexagonal close-packed nanosphere ($D = 500$ nm) monolayers on glass and Si substrates ($1 \times 1$ cm$^2$), which were prepared by an air/water interface method[23,63–65], were used as a template for electron beam deposition. For pure Pd NP samples, the substrates were coated by Pd with a varied thickness of $t_{Pd}$, under a constant deposition rate of 0.05 nm/s, and the sample holder rotated azimuthally with a constant rotation rate of 30 rpm during deposition process. For Pd-Ag, Pd-Au or Pd-Co composite NP sample, materials were placed in two independent crucibles on two sides of the chamber, and the vapor incident angles to substrate normal were 10° and −10°, respectively. The deposition rates and thicknesses of Pd and Ag/Au/Co were monitored independently by two separated quartz crystal microbalances (QCM). By controlling the deposition rates of Pd and Ag/Au/Co, $Pd_{80}Ag_{20}$, $Pd_{80}Au_{20}$, and $Pd_{80}Co_{20}$ NPs were realized.

**Morphology and composition characterization**. Scanning electron microscopy (SEM) was performed with a Thermo Fisher Scientific (FEI) Teneo field emission scanning electron microscope (FESEM). Energy-dispersive spectroscopy (EDS) elemental mapping was performed with 150 mm Oxford XMaxN detector. Ultra-high-resolution SEM was performed with a SU-9000, Hitachi (with resolution of 0.4 nm at 30 kV).

**Hydrogen sensing measurement**. All optical isotherm, LOD, response/release time measurements are performed in a home-made vacuum chamber with two quartz windows[23]. The hydrogen pressure was monitored by three independent pressure transducers with different which cover the pressure range of $10^{-6}$ to 1.1 bar (two PX409-USBH, Omega and a Baratron, MKS). Optical transmission measurements were performed with an unpolarized collimated halogen lamp light source (HL-2000, Ocean Optics) and a spectrometer (USB4000-VIS-NIR-ES, Ocean Optics). The optical response/release time measurements were performed at 12.5 Hz sampling frequency (4 ms integration time with 10 averages). The LOD measurements were performed at 1.25 Hz sampling frequency (4 ms integration time with 100 averages), and $\Delta T/T$ responses were averaged over wavelength range of $\lambda = 500$–660 nm for the best SNR. For LOD measurements in flow mode, ultra-high purity hydrogen gas (Airgas) was diluted with ultra-high purity nitrogen gas (Airgas) or synthetic gas (Airgas) to targeted concentrations by commercial gas blenders (GB-103, MCQ Instruments). The gas flow rate was kept constant at 400 ml/min for all measurements. All experiments (except the temperature-dependent experiments) were performed at constant 25 °C.

**FDTD calculations**. FDTD calculations of Pd hydride NP samples were carried out using a commercial software (Lumerical FDTD Solutions)[66]. The geometric parameters of Pd cap were obtained from MATLAB simulation. The mesh size of $2 \times 2 \times 2$ nm was chosen. The refractive index of glass and PS was chosen to be 1.5 and 1.59, respectively, and the optical parameters of Pd and PdH$_x$ were extracted from ref. [67].

**PMMA coating**. PMMA (Sigma Aldrich, 10 mg/ml dissolved in acetone by heating up the mix to 80 °C and cooling down to the room temperature, $M_w = 15,000$) was spin-coated on sensors at 5000 r.p.m. for 120 s followed by a soft baking at 85°C on a hotplate for 20 min. Using the same coating process on a clean glass substrate results in a ~50 nm PMMA film, as measured by an atomic force microscope (NX-10, Park Instrument).

## Data availability

The data that support the finding of this study are available from the corresponding authors upon reasonable request.

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

## Acknowledgements

The authors thank Prof. Yiping Zhao for his generosity in sharing his nano-fabrication tools with us. This work was supported by Savannah River National Laboratory's Laboratory Directed and Development program (SRNL is managed and operated by Savannah River Nuclear Solutions, LLC under contract no. DE-AC09-08SR22470). M.H.P. acknowledges support from the U.S. Department of Energy, Office of Basic Energy Sciences, Division of Materials Sciences and Engineering under Award No. DE-FG02-07ER46438 and the University of South Florida Nexus Initiative (UNI) under Award No. R15301.

## Author contributions

H.M.L. designed and fabricated the samples, performed the sensing experiments, performed FDTD simulations, analyzed the experimental data, and wrote the first draft of the paper. M.T.P. and T.G. co-wrote and edited the paper. R.P.M. and M.H.P. performed SEM measurements and edited the paper. G.K.L. analyzed the experimental data, co-wrote the paper, and supervised the project. T.D.N. was responsible for project planning, group managing, and manuscript writing. All authors gave feedback on the final paper.

## Competing interests

The authors declare no competing interests.
