## [Peer Review File · Nature Communications]

Reviewers' Comments:

Reviewer #1:

Remarks to the Author:

Luong et al. report optical hydrogen sensors that are comprised of what they call "nano-patchy particle arrays" and that are characterized by ppm limit of detection and sub-second response. At the general level, I find their study interesting, well presented and also scientifically sound. Furthermore, the reported sensor metrics are impressive and the concept of the nano-patchy arrays is very interesting. However, before I can recommend publication in Nature Communications, the authors need to address fully and in detail the following points:

- The name "nano-patchy particle arrays", which is introduced in the title, is not very appealing since it – to me – does not say anything. In other words, it does not really describe the used structures. I would strongly recommend the authors to reconsider their choice.
- Related to the term "nano-patchy", one of the key shortcomings of this work is that no detailed structural characterization of the Pd, PdAg and PdCo layers evaporated onto the polystyrene sphere monolayer is provided. This is, however, critical to be able to fully understand the sensor performance since it is well known that ultrathin metal layers in the few nm thickness regime may form isolated islands (=small nanoparticles) rather than a continuous film. Here the authors describe their results from the perspective of the formation of films as thin as 2nm without providing explicit evidence that this is correct. The SEM images do not have high enough resolution. I thus recommend imaging in a transmission electron microscope with higher spatial resolution.
- The authors state that in alloys, grain boundaries and dislocations are formed due to lattice mismatch between alloyants. I would not a priori agree with this statement since any metal film, if not grown epitaxially, will be polycrystalline (=have grain boundaries) and dislocation rich as a consequence of the nucleation and growth process. Therefore, also in pure Pd grain boundaries and dislocations will be present (see e.g. the works by Ulvestad et al. *Nat Mater* 2017, 16(5): 565-571; *Nature Communications* 2017, 8(1): 1376; *ACS Nano* 2017, 11(11): 10945-10954)
- The authors state that spectral-shift based plasmonic sensors need "sophisticated fittings and create non-specific response time readout". On the first point, I think calling the typically used procedures "sophisticated" is a bit exaggerated, since it typically entails fitting of polynomial or Lorentzian functions, which is straightforward (see e.g. Dahlin AB, et al. *Analytical Chemistry* 2006, 78(13): 4416-4423.). On the second point, I do not understand what the authors mean with "non-specific response time readout". Finally, it is not correct that the fitting procedures rally are necessary since it has been demonstrated that single-wavelength readout actually even may improve the detection limit (Wadell C, et al. *Nano Letters* 2015, 15(5): 3563-3570.).
- The authors claim in the introduction that their sensor has the highest sensitivity reported to date. This statement is wrong and needs to be adjusted and put into perspective throughout this work. There are reports of hydrogen sensors with detection limits in the ppb range, for example: Lupan, O., et al. *Sensors Actuators B Chem.* 2018, 254, 1259–1270. Lee, et al. *Nanoscale* 2015. Xiao, M.; et al. *ACS Sensors* 2018, 3 (4), 749–756.
- The authors state that it is counter intuitive that hysteresis is reduced and plateau slope increases upon decreasing tPd. In my opinion this is very much expected and in good agreement with observations for nanoparticles (Yamauchi M, et al. *Journal of Physical Chemistry C* 2008, 112(9): 3294-3299; Langhammer C, et al. *Chemical Physics Letters* 2010, 488: 62-66.; Syrenova S, et al. *Nature Materials* 2015, 14: 1236–1244) and relates to the magnitude of the strain-induced energy barrier for hydride formation and decomposition. Therefore, this comment is strongly related to my first one, where I ask for more detailed structural characterization of the Pd (and alloy) thin films to ensure that they are films and not small particles.
- On page 7 it is stated that a peak in the t50 vs pH₂ relation is characteristic of Pd-based hydrogen sensors. While this observation is generally new to me, if it really is characteristic for Pd, what is its origin?
- The authors analyze their data with respect to the volume-to-surface ratio of their structures (VSR) but, and make an attempt of explaining how this number is derived in the supporting

information (Figs S1 and S2). However, to me it is not clear how these numbers are derived and more clear explanation is necessary as this is a key point of the paper.

- The authors argue that their thinnest films due to their optimized VSR exhibits the best performance in terms of detection limit and response time. While I agree with this conclusion, I think it is very important from an application perspective to evaluate the long terms stability, as well as reproducibility of the ultrafast and ultrasensitive response over time and multiple (de)hydrogenation cycles. I am afraid that such thin films are very fragile and that their behavior may be significantly altered upon cycling.
- The authors have opted to select PdAg alloys rather than PdAu alloys which have been studied much more extensively. It would therefore be interesting to know the reason for this choice and also it would be helpful to provide a direct comparison with a PdAu system to see how much of the enhanced performance observed is due to the fact that the sensors are designed in this "nano-patchy" fashion and how much is intrinsic to the PdAg system.
- The noise level in the raw data for Pd80Ag20 and Pd80Co20 presented in Figure S16, the noise level for the Co-alloy system appears a lot higher. Why? Furthermore, the authors define their detection limit as 3sigma but the actual level of 3 sigma is not included in the figure and it is therefore not possible to check whether the made LoD claim is correct. In particular for the Co system I am not sure if a LoD of 2.5 ppm really can be claimed.
- The authors compared "vacuum-mode and flow-mode sensing" and find similar sensing response in terms of LoD. Why would they not? Also in flow mode only inert gas background is used. It would, however, from an application perspective be very interesting to see how their sensors perform at more realistic conditions like in air, and how that affects the LoD.
- In the conclusions section the authors claim a "groundbreaking level of sensitivity" and that their "sensors demonstrate viable path forward to spark-proof optical sensors for hydrogen detection applications that no other method has been proven to date". These are very strong statements which, in particular for the first one, are simply not correct. I would therefore strongly advise the authors to not oversell their work in this way since it is absolutely not necessary. Their work is good enough as it is.

Reviewer #2:

Remarks to the Author:

- What are the major claims of the paper?

This paper claims to have created optical hydrogen sensors with a 2.5 ppm limit of detection with a response time of 0.85 second at 1 mbar by utilizing novel nanostructures to increase the surface to volume ratio of the metal hydride. By investigating different deposition thicknesses and angles, as well as different alloying combinations, they were able to optimize their sensor.

- Are the claims novel? If not, please identify the major papers that compromise novelty

This proposed nano-achitecture is novel to my knowledge.

- Will the paper be of interest to others in the field?

Yes.

- Will the paper influence thinking in the field?

Yes, this novel nanostructure could inspire more sensors similar to its design.

Questions:

1. In Figure 2, you should show the data for $t_{Pd} = 0$ (bare PS beads) for each subplot if you are going to explicitly refer to it in the paper.
2. Why is there a peak in 2f? Literature is cited to show that this peak is not unusual, but is the physical cause known for this peak?
3. Is there a reason why the authors chose PdAg and PdCo as their test alloys instead of the more common PdAu? What results would they expect from experiments using PdAu with their nanostructure? Along with this, why did the authors choose 80/20 alloy ration over other options? Recent work on Pd alloys for hydrogenation should be cited, e.g. Palm et al, ACS Applied Materials & Interfaces 11, 45057-45067 (2019) and Bannenberg, et al, ACS Appl. Mater. Interfaces 2019,

11 (17), 15489–15497.

4. How do the authors determine the start time of hydrogen exposure? Is it defined as the time that the gas is switched over, or is there any accounting for gas change over time within their chamber?

5. Line 50, what limit of detection are you drawing the line at? Nugroho et al. Nat. Mater. 18, 489–495 (2019) claims a $t_{90} = 1$ sec sensing time at 1 mbar exposure with a LOD of 3 ppm in an Ar atmosphere.

Reviewer #3:

Remarks to the Author:

The authors reported on a compact optical hydrogen sensing platform, which was comprised of Pd and Pd-alloy nano-patchy arrays. The hexagonally packed nanoscale arrays were fabricated by single-metal glancing-angle deposition on polystyrene (PS) nanosphere monolayers. This fabrication process was simple and with no post-processing required. The tunable film thicknesses, patchy diameters and shapes (hemispheres or donuts) resulted in tunable and rapid response rates. By alloying with Co, the response time of the metasurfaces was reduced below 0.85 s from 1-100 mbar of H₂ partial pressure, surpassing the strictest requirements for H₂ sensing while preserving good accuracy (<2.5% full scale) and 2.5-ppm limit of detection (LOD).

The demonstrations of the hydrogen sensing performances of Pd and Pd-alloy nanostructure arrays have been reported in many previous works. The detection limits of hydrogen concentrations can be down to 100 ppm (ACS Sens. 2020, 5, 4, 917), and PdAg nanocomposite exhibits a high sensitivity of 5.5% for 1.6% H₂ concentration (ACS Sens. 2020, 5, 2367). The experiments were carefully performed and the results were convincing. However, although the hydrogen sensing performance was enhanced, the measurements were only performed in a lab experiment manner. It is far away from the practical application. I do not think the novelty or the importance of this work is sufficient for Nature Communications. It is more suitable for a journal on nanoscience or materials.

[1] What are the diameters of the PS nanospheres in Figure 1? Why were the PS nanospheres with such diameters chosen?

[2] What are the morphologies of the pure Pd NP samples simulated in Figure 3a? What does the scale bar stand for?

[3] The top-view SEM images and corresponding simulated morphologies of the pure Pd NP samples fabricated with a vapor incident angle of $\theta = 50$ degrees should be provided in Figure 3a.

[4] The optical hydrogen absorption spectrum, sensor accuracy and response time of the Pd₈₀Co₂₀ ($t = 15$ nm) should be provided in Figure 4 for comparison.

[5] The authors are suggested to give a standard curve of the optical response as a function of the hydrogen concentration.

[6] The authors investigated the optical response of the Pd-alloy nano-patchy arrays in vacuum and nitrogen. They did not consider the disturbances to the hydrogen sensing in practical situations, such as the influences from temperature, moisture, and other gases (e.g., CO, CO₂, air, etc.).

[7] The error bar should be included in Figure 5c.

Response Letter:

(We thank all the reviewers for their time and useful comments that helped improved the manuscript substantially. We have revised our manuscript in light of these comments. Below are our detailed responses to reviewer's each comment)

Responses to Comments from Reviewer #1

Luong et al. report optical hydrogen sensors that are comprised of what they call “nano-patchy particle arrays” and that are characterized by ppm limit of detection and sub-second response. At the general level, I find their study interesting, well presented and also scientifically sound. Furthermore, the reported sensor metrics are impressive and the concept of the nano-patchy arrays is very interesting. However, before I can recommend publication in Nature Communications, the authors need to address fully and in detail the following points:

Author Reply: We thank the reviewer for the positive assessment of our work. We are happy to have an opportunity to address the reviewer's critical remarks, important questions, and insightful suggestions.

1. The name “nano-patchy particle arrays”, which is introduced in the title, is not very appealing since it – to me – does not say anything. In other words, it does not really describe the used structures. I would strongly recommend the authors to reconsider their choice.

Author Reply: Following the reviewer's recommendation, we have changed the title to: “*Sub-second and ppm-level Optical Sensing of Hydrogen Using Templated Control of Nano-hydride Geometry and Composition*”

2. Related to the term “nano-patchy”, one of the key shortcomings of this work is that no detailed structural characterization of the Pd, PdAg and PdCo layers evaporated onto the polystyrene sphere monolayer is provided. This is, however, critical to be able to fully understand the sensor performance since it is well known that ultrathin metal layers in the few nm thickness regime may form isolated islands (=small nanoparticles) rather than a continuous film. Here the authors describe their results from the perspective of the formation of films as thin as 2nm without providing explicit evidence that this is correct. The SEM images do not have high enough resolution. I thus recommend imaging in a transmission electron microscope with higher spatial resolution.

Author Reply: We thank the reviewer for these insightful comments. We have characterized the Pd NP structure with an *ultrahigh-resolution* (0.4 nm at 30 kV) scanning electron microscope (SEM) (SU-9000, Hitachi) (Figure R1).

Figure R1. (a) Scanning electron microscopy (SEM) images showing morphology differences of Pd NP with different deposited thicknesses (t_{Pd} ranges from 1.5 to 15 nm). Scale bar: 100 nm. Magnified views (150 nm \times 150 nm) of SEM images in Figure R1a can be found in Figure 1d. (b) Cross-sectional SEM image and EDS elemental maps of NP_5^{50} showing a metal cap formed on the top surface of a PS nanosphere. Scale bar: 500 nm.

The morphology of $NP_{1.5}^{50}$ (NP_t^θ , where θ ($^\circ$) and t (nm) represent the incident angle of the metal vapor and the thickness of the deposited film, respectively) contains many sub-10-nm granules, and these granules fully cover the top surface of the polystyrene nanosphere. The size of the granules grows when $t_{Pd} = 2$ nm, and coalescence via a neck (or bridge) connection between neighboring clusters¹ can be noticed at the thickness of $t_{Pd} = 3.5$ nm. A continuous film is formed at a thickness of 5 nm, as these bridge connections are successively grown (a cross-sectional SEM image and EDS elemental maps of NP_5^{50} can be found in Figure R1b). Once the continuous film is shaped at a thickness of 5-nm, another

distinct coalescent mechanism is observed: the size of the clusters increases as the thickness increases, associated with the reduction of the cluster density.

Figure R2. Scanning electron microscopy (SEM) images showing morphology differences of NP with different deposited thicknesses (5 and 15 nm) and compositions. Scale bar: 100 nm.

We find that while $\text{Pd}_{80}\text{Co}_{20}$ and $\text{Pd}_{80}\text{Au}_{20}$ samples show a similar evolution of morphology as Pd samples (*i.e.* the size of the clusters increases as the thickness increases from 5 to 15 nm, accompanied with the reduction of the cluster density), $\text{Pd}_{80}\text{Ag}_{20}$ samples exhibit a very different behavior. The $\text{Pd}_{80}\text{Ag}_{20}$ layer does not spread out and cover fully the top surface of the nanosphere, but forms several islands with different sizes and shapes. We attribute this difference to a low sticking coefficient and large surface diffusion coefficient of Ag, which encourages nucleation of Ag, during the vacuum deposition.²⁻³

In order to quantify these observations, we define a volume fraction, V_f , such that

$$V_f = V_{dep}/V_{meas} \quad (\text{R1})$$

$$V_{dep} = \frac{4\pi}{6} \left((R + t_{dep})^3 - R^3 \right) \quad (\text{R2})$$

$$V_{meas} = \frac{4\pi}{6} \left((R + t_{meas})^3 - R^3 \right), \quad (\text{R3})$$

where V_{dep} is the hemispherical volume of material deposited based on a deposition of material thickness, t_{dep} , (as determined by the quartz crystal microbalance during the fabrication) onto a sphere of radius, R ($R = 500$ nm in this case). V_{meas} would be the volume of a hemispherical shell on a sphere of radius R if the thickness of the shell, t_{meas} , matched the average grain size measured in the high-resolution SEM images (see Figure S8 for the grain size analysis). Figure R3a shows the calculated V_f for the different Pd films, and while the standard deviation, σ , is rather large for the calculations, a general trend can be observed. Consistent with the qualitative description above, films with $t_{Pd} \geq 3.5$ nm have a volume fraction that exceeds the percolation threshold p_c for bond percolation ($p_c = 0.347296$), and films with $t_{Pd} \geq 5$ nm have a volume fraction that exceeds p_c for site percolation ($p_c = 0.5$).⁴ In this case, we consider a film to be island-like if $V_f < 0.5$ and film-like if $V_f \geq 0.5$.

As shown in Figure R2, different alloying elements (Ag, Au, Co) impact the nanoparticle morphology differently. Therefore, it is also interesting to compare the effects of the different alloying elements on V_f . Figure R3b presents the V_f for the different Pd alloy films. The general trend for V_f is $\text{PdAg} < \text{Pd} < \text{PdAu} < \text{PdCo}$, with $\text{Pd} \approx \text{PdAg}$ and $\text{PdAu} \approx \text{PdCo}$ more loosely. It is worth noting that these films all have $V_f \geq 0.5$, but also have large σ values, where Pd and PdAg have the largest variations.

Figure R3. Calculated volume fraction, V_f , of (a) the Pd samples versus deposited thickness and (b) the different Pd alloys for $t_{dep} = 5$ and 15 nm. The values of standard deviation, σ , for the V_f calculations are illustrated by the pink squares and placed above the bars in (a) and (b), respectively.

Change made to the manuscript:

- Magnified views of SEM images in Figure R1 have been added to the main text as Figure 1d, and a paragraph describing this figure has been added to Page 4, Line 83.
- Figure R1b has been added to the main text as Figure 1e.
- Figures R1a, R3, and R2 have been added to the SI as Figure S8, S9, and S10, respectively. Morphological transitions of Pd and Pd-composite NPs with different deposited thicknesses have been discussed throughout in Subsection S2.2 in the SI.

3. The authors state that in alloys, grain boundaries and dislocations are formed due to lattice mismatch between alloyants. I would not a priori agree with this statement since any metal film, if not grown epitaxially, will be polycrystalline (=have grain boundaries) and dislocation rich as a consequence of the nucleation and growth process. Therefore, also in pure Pd grain boundaries and dislocations will be present (see e.g. the works by Ulvestad et al. Nat Mater 2017, 16(5): 565-571; Nature Communications 2017, 8(1): 1376; ACS Nano 2017, 11(11): 10945-10954).

Author Reply: We thank the reviewer for the insightful comments and for referring us to the useful papers. We agree with the reviewer that such grain boundaries and dislocations also appear in our polycrystalline Pd metal films. It appears that more imperfections are likely present in the alloy films. Such imperfections in the alloys likely compensate the lattice strain upon (de)hydrogenation, decrease the abrupt and large volume expansion occurring in smaller mixed phase regions leading to smaller strain-induced energy barrier, suppress the structural deformation, thus improving the sensor's stability and response rate as well.⁵⁻⁸

Change made to the manuscript:

We have revised the sentences in Page 2, Line 42, which now read:

“It has been believed that the enhanced reaction kinetics in the alloying metal hydrides is associated with the reduction of the enthalpy of formation due to the reduced abrupt volume expansion occurring in smaller mixed phase regions, resulting in a reduction of energy barrier for hydride formation and dissociation, and the improved diffusion rate upon (de)hydrogenation.”^{12,15,17-20}

4. The authors state that spectral-shift based plasmonic sensors need “sophisticated fittings and create non-specific response time readout”. On the first point, I think calling the typically used procedures “sophisticated” is a bit exaggerated, since it typically entails fitting of polynomial or Lorentzian functions, which is straightforward (see e.g. Dahlin AB, et al. Analytical Chemistry 2006, 78(13): 4416-4423.). On the second point, I do not understand what the authors mean with “non-specific response time readout”. Finally, it is not correct that the fitting procedures rally are necessary since it has been demonstrated that single-wavelength readout actually even may improve the detection limit (Wadell C, et al. Nano Letters 2015, 15(5): 3563-3570.).

Author Reply: We thank the reviewer for pointing this out. After studying the suggested paper carefully, we agree with the reviewer and have removed that statement. The introduction part has been revised accordingly.

Change made to the manuscript:

We have removed the sentences in Page 3, Line 55.

5. The authors claim in the introduction that their sensor has the highest sensitivity reported to date. This statement is wrong and needs to be adjusted and put into perspective throughout this work. There are reports of hydrogen sensors with detection limits in the ppb range, for example: Lupan, O., et al. Sensors Actuators B Chem. 2018, 254, 1259–1270. Lee, et al. Nanoscale 2015. Xiao, M.; et al. ACS Sensors 2018, 3 (4), 749–756.

Author Reply: We agree with the reviewer and have removed this statement in the revised manuscript.

Change made to the manuscript: We have removed the statement in Page 3, Line 60.

6. The authors state that it is counter intuitive that hysteresis is reduced and plateau slope increases upon decreasing t_{Pd} . In my opinion this is very much expected and in good agreement with observations for nanoparticles (Yamauchi M, et al. Journal of Physical Chemistry C 2008, 112(9): 3294-3299; Langhammer C, et al. Chemical Physics Letters 2010, 488: 62-66.; Syrenova S, et al. Nature Materials 2015, 14: 1236–1244) and relates to the magnitude of the strain-induced energy barrier for hydride formation and decomposition. Therefore, this comment is strongly related to my first one, where I ask for more detailed structural characterization of the Pd (and alloy) thin films to ensure that they are films and not small particles.

Author Reply: We thank the reviewer for the valuable comments. The transition in optical behavior occurs for $t_{Pd} < 5$ nm, which matches the transition between island-like to film-like morphologies described in Comment #1. We have revised intensively the Subsection S2.3 in SI, where the size dependences of $\ln(P_{Abs}/P_{Des})$, P_{Abs} , and P_{Des} in $NP_{t_{Pd}}^{50}$ films are discussed throughout.

Change made to the manuscript:

- Subsection S2.3 in SI has been revised based on the morphological transitions of the Pd thin film.
- We have revised the sentences in Page 5, Line 112, which now read:

“The mixed region hysteresis is reduced, and the slope of the plateau pressure increases with decreasing t_{Pd} . The size dependences of the plateau slope and hysteresis have been previously observed in Pd nanoparticles with a narrow size distribution, but we cannot discount the effect of size distribution on the slope for $NP_{t_{Pd}}^{50}$ samples (Supplementary Figs. S8–9).³⁰”

- This sentence has been added in Page 6, Line 120:

“It is worthwhile to note that this critical thickness corresponds well with the expected transition from a separated island-like morphology to a percolating film morphology”

7. On page 7 it is stated that a peak in the t_{50} vs p_{H_2} relation is characteristic of Pd-based hydrogen sensors. While this observation is generally new to me, if it really is characteristic for Pd, what is its origin?

Author Reply: The peak in the t_{90} versus P_{H_2} relation for P_{Abs} has been reported in some Pd-based hydrogen sensor,⁹⁻¹¹ however, the origin of this observation has not been discussed. We hypothesize that the peak is associated to the abrupt change in hydride volume during the $\alpha - \beta$ phase transition. Such abrupt change or sudden entropy change occurs only if the hydride system thermodynamically overcomes the large strain-induced energy barrier upon hydrogenation, which requires longer time to reach the equilibrium. The strain-induced energy barrier is relatively strong in pure Pd based sensors, which results in the large t_{90} peak in this pressure regime. It is clear in Figure 2f that such barrier is larger with a larger grain size and/or a thicker film thickness,⁵ and interestingly, the barrier becomes much less significant as the structure transforms from film-like to nanoparticle island-like for $t_{Pd} \leq 2$ nm.

Change made to the manuscript: We have added the above discussions in Page 7, Line 144.

8. The authors analyze their data with respect to the volume-to-surface ratio of their structures (VSR) but, and make an attempt of explaining how this number is derived in the supporting information (Figs S1 and S2). However, to me it is not clear how these numbers are derived and more clear explanation is necessary as this is a key point of the paper.

Author Reply: We thank the reviewer for the comment. We estimate the thickness distribution, surface, and volume of the hemisphere cap, based on a simple simulation including a uniform vapor flux approaches in direction $\hat{l}(\theta_0, \varphi_0)$ to an array of hexagonal close-packed nanosphere with diameter $D = 500$ nm (Figure R4a). The thickness distribution of the nanosphere O (highlighted in orange) is calculated by considering the shadowing effects from 36 nearest-neighbor nanospheres. The surface of nanosphere O is broken down into smaller surface elements, and each of them is labelled by the polar coordinate of its center (θ, φ) , where $\theta = i\Delta\theta$, $\varphi = j\Delta\varphi$, $\Delta\theta = \Delta\varphi = 0.5^\circ$, $i = 0, 1, \dots, 360, j = 0, 1, \dots, 720$.

Figure R4. (a) A cartoon illustrates the vapor deposition on an array of nanospheres. (b) The surface of nanosphere O split into several surface elements. (c) A surface element on the nanosphere O .

Our simulation is based on these following assumptions:

- Only the shadowing effect and material accumulation are considered. Other physical processes, such as surface diffusion or material penetration, are neglected.
- The deposition at different surface elements happens simultaneously as long as they are directly exposed to vapor.
- The as-deposited film is non-porous and is uniform within each surface element.

In our experimental metal deposition, the substrate holder was rotated azimuthally at a constant rate of 30 rpm to thoroughly cover the top surface of nanosphere. Therefore, in order to mimic this process, we break down the simulation into 3600 steps. We start at $\varphi_0 = 0^\circ$ (step index $k_i = 1$), a 0.1° azimuthal rotation of $\hat{l}(\theta_0, \varphi_0)$ happens at the end of each step, until a round of azimuthal rotation is completed ($\varphi_0 = 359.9^\circ$) (step index $k_f = 3600$). In each simulation step, the thickness at each surface element $h(\theta, \varphi)$ is updated,

$$h_{k+1}(\theta, \varphi) = h_k(\theta, \varphi) + \Delta h_k.$$

The change of thickness for each step Δh_k is determined by whether surface element (θ, φ) is directly exposed to the vapor flux or not:

- If the surface element (θ, φ) is under the shadow of other structures (deposited materials on neighboring bead in the previous steps are also considered), then $\Delta h_k = 0$.

- If the surface element (θ, φ) can receive vapor, then $\Delta h_k = \frac{\Delta m}{\rho \cdot S(\theta, \varphi)}$, where Δm is the mass of material deposited on surface element (θ, φ) within time Δt , ρ is the density of the material, and $S(\theta, \varphi)$ is the area of surface element (θ, φ) .

The vapor flux Φ is defined as,

$$\Phi = \frac{\Delta m}{\Delta t \cdot S_N} = \text{const},$$

where S_N is the projection of $S(\theta, \varphi)$ onto the plane perpendicular to the vapor flux $\hat{l}(\theta_0, \varphi_0)$. With β is the angle between \hat{l} and the surface normal vector \hat{n} (Figure R4c), S_N can be written as,

$$S_N = S(\theta, \varphi) \cos\beta.$$

Combining these relations, we achieve,

$$\Delta h_k = \frac{\Phi}{\rho} \Delta t \cos\beta.$$

For the simulation of $\text{NP}_{t_{\text{Pd}}}^{\theta_0}$ sample, in each step k , we set $\frac{\Phi}{\rho} \Delta t = \frac{t_{\text{Pd}}}{k_f} = \frac{t_{\text{Pd}}}{3600}$ (nm). After the last simulation step of k_f , a 3-dimension hemisphere cap are rendered based on the thickness distribution on the nanosphere O , and the volume and surface area of the patchy particle are simply obtained by using double and triple integral built-in function of MATLAB. The results are presented in Figures S2 and S3 in SI.

Change made to the manuscript: We have added the above discussions in SI as Subsection S1.1.

9. The authors argue that their thinnest films due to their optimized VSR exhibits the best performance in terms of detection limit and response time. While I agree with this conclusion, I think it is very important from an application perspective to evaluate the long terms stability, as well as reproducibility of the ultrafast and ultrasensitive response over time and multiple (de)hydrogenation cycles. I am afraid that such thin films are very fragile and that their behavior may be significantly altered upon cycling.

Author Reply: We agree with the reviewer that it is very important to evaluate the long-term stability and reproducibility over time and multiple (de)hydrogenation cycles for sensing application. In this test, we focus on the $\text{Pd}_{80}\text{Co}_{20} \text{NP}_5^{50}$ sample, which is a potential candidate for an ultrafast and ultrasensitive hydrogen sensor.

For a long-term stability assessment, we simply placed $\text{Pd}_{80}\text{Co}_{20} \text{NP}_5^{50}$ sample in the ambient condition (temperature 22-23°C, relative humidity (RH) 35%) and characterize the response time and limit of detection (LOD) of the sensor in a weekly basis (week 1 to 6). These obtained results are then compared to a 10-month old sample, in Figure R5.

The $\text{Pd}_{80}\text{Co}_{20} \text{NP}_5^{50}$ sample shows a good stability without the sign of degradation, upon >300 of (de)hydrogenation cycles. The first 100 cycles (1/1 minute of loading/unloading with 2% H_2 in synthetic gas) of 3-weeks old and 10-months old sample are summarized in Figure R5a. While a noticeable

reduction of sensor signal due to aging can be seen in the 10-months old sample (Figure R5b), upon cycling, the signal is recovering back to that of a fresh sample.

Figure R5. (a) $\Delta T/T$ response of $Pd_{80}Co_{20} NP_5^{50}$ (3-weeks and 10-months old) upon 100 cycles (1/1 minute of loading/unloading) of 2% H_2 in synthetic gas (400 ml/min) and (b) $\Delta T/T$ response normalized to the response of a fresh sample, obtained in the same condition. The error bars denote the standard deviation from 100 cycles. (c) Long-term $\Delta T/T$ response of $Pd_{80}Co_{20} NP_5^{50}$ with different hydrogen concentrations (C_{H_2}) of 100 – 2.5 ppm, measured in flowing nitrogen (400 ml/min). Shaded areas denote the periods where the sensor is exposed to hydrogen. (d) Long-term response time of $Pd_{80}Co_{20} NP_5^{50}$ with 1-100 mbar pure hydrogen pulse. (e) Measured $\Delta T/T$ response as a function of P_{H_2} in vacuum/pure hydrogen (solid symbol) and C_{H_2} in flowing nitrogen (400 ml/min) (half-up filled symbols). Inset: the blue dashed line denotes the defined LOD at $3\sigma \approx 0.011\%$ ($\sigma = 0.0035\%$, is the noise of the acquired signal with N_2 carrier air, see Section S9 in SI).

The sign of the aging effect can be seen by the degraded performances of the sensor over the times. In particular, the response time t_{90} (at $P_{H_2} = 1 - 100$ mbar) increases about ~ 2 times and ~ 3 times after a period of 4-weeks and 10-months, respectively (Figures R5c). In addition, the long-term $\Delta T/T$ responses show a significant reduction both in vacuum mode and flow mode (Figures R5d and 5e), just after 4-5 weeks in air. We ascribed the degradation of the sensor to a small trace amount of poison gases exist in the ambient air, (e.g. CO) (see the deactivation test with CH_4 , CO_2 , and CO in Comment #6, Reviewer #3). However, we note that the response time and LOD of the sample still are < 2.5 s (at 1 mbar) and < 10 ppm, respectively, over a > 10 -month period. The degradation of the $Pd_{80}Co_{20} NP_5^{50}$ sample upon long-

term storage in air can affect the accuracy, sensitivity, and response time of the sensor. Hence, we look for a solution to mitigate this process. Inspired by the work of Nugroho *et al.*,¹² we coated the Pd₈₀Co₂₀ NP₅⁵⁰ sample with a ~50-nm layer of polymethyl methacrylate (PMMA) (by spin-coating of PMMA dissolved in acetone, more details can be found in Methods section in the main-text), which has been demonstrated to effectively block the poisonous species. In this test, we store the PMMA-coated and uncoated samples in an identical condition and measure their sensing metrics.

Figure R6. (a) $\Delta T/T$ response of Pd₈₀Co₂₀ NP₅⁵⁰/PMMA (3-weeks old) upon 150 cycles (1/1 minute of loading/unloading) of 2% H₂ in synthetic gas (400 ml/min). (b) Long-term $\Delta T/T$ response of Pd₈₀Co₂₀ NP₅⁵⁰/PMMA with different hydrogen concentrations (C_{H_2}) of 100 – 2.5 ppm, measured in flowing nitrogen (400 ml/min). Shaded areas denote the periods where the sensor is exposed to hydrogen. (c) Long-term response time of Pd₈₀Co₂₀ NP₅⁵⁰/PMMA with 1-100 mbar pure hydrogen pulse. (d) Measured $\Delta T/T$ response as a function of P_{H_2} in vacuum/pure hydrogen (solid symbol) and with different C_{H_2} in flowing N₂ (half-up filled symbols). Inset: the blue dashed line denotes the defined LOD at $3\sigma \approx 0.011\%$ ($\sigma = 0.0035\%$, is the noise of the acquired signal with N₂ carrier air, see Section S9 in SI).

The sensing performances of Pd₈₀Co₂₀ NP₅⁵⁰/PMMA sensor are summarized in Figure R6. After storing in air for 6-weeks and underwent >200 cycles of (de)hydrogenation with 2% H₂, we observe very little variances in response time over the pressure range of 1-100 mbar (t_{90} are <1.5 s (at 1 mbar)) and insignificant reduction of sensor signal upon exposure to pulses of very low H₂ concentration (100 – 2.5 ppm). Clearly, the degradation of the sensor performance is significantly slowed-down with a polymer coating layer. In addition, we also demonstrated that this PMMA coating can efficiently block the poisonous gas species (e.g. CO, CO₂, and CH₄) as well as reduce the influences from moisture (see Comment #6, Reviewer #3). This is an interesting improvement for this hydrogen sensing platform that stimulates further study.

Change made to the manuscript:

- The above discussions have been added in SI as Section S10.
- We have added a paragraph in main-text, at Page 15, Line 319, to discuss about the reproducibility of the ultrafast and ultrasensitive response over time and multiple (de)hydrogenation cycles, and deactivation tests with disturbance gases.
- The abstract and conclusion have been revised accordingly.

10. The authors have opted to select PdAg alloys rather than PdAu alloys which have been studied much more extensively. It would therefore be interesting to know the reason for this choice and also it would be helpful to provide a direct comparison with a PdAu system to see how much of the enhanced performance observed is due to the fact that the sensors are designed in this “nano-patchy” fashion and how much is intrinsic to the PdAg system.

Author Reply: We thank the reviewer for raising this interesting question. We used PdAg, instead of PdAu, simply due to the availability of the materials for electron beam evaporation at the time of the sample fabrication.

We agree that having a PdAu system would be very helpful, as it allows for a direct comparison to existing works using the same materials. To address this question, we have fabricated Pd₈₀Au₂₀ NP₅⁵⁰ and NP₁₅⁵⁰ samples using the similar fabrication processes for PdAg and PdCo systems. The high-resolution SEM images and energy-dispersive X-ray spectroscopy (EDS) elemental mapping, which are shown in Figures R7a and R2, indicate the desired composition and structure of these samples.

Figure R7. (a) SEM and EDS images of Pd₈₀Au₂₀ ($t = 5$ nm) composite NP samples. Scale bar corresponds to 500 nm. A table shows the elemental atomic composition (at. %) of the Pd₈₀Au₂₀ NP, which is consistent with the desired composition; (b) Experimental optical transmission spectra $T(\lambda)$ and (c) optical transmission changes $\Delta T(\lambda) = T_{1000 \text{ mbar}} - T_{0 \text{ mbar}}$ of PdAg, PdCo, and PdAu samples with different deposited thicknesses;

(d) Optical hydrogen sorption isotherm of composite NP, extracted at $\Delta T(\lambda)$ maxima. Colored arrows indicate sorption direction. (e) Sensor accuracy of composite NP sensors at specific normalized ΔT readout over hydrogen pressure range of 10^1 μbar to 10^6 μbar . (f) Response time of NP sensors with pulse of hydrogen pressure from 1 mbar to 100 mbar.

The hydrogen sorption characteristics of the $\text{Pd}_{80}\text{Au}_{20}$ alloy NP system are examined as previously. The optical transmission spectra ($T(\%)$) of PdAg, PdCo, and PdAu samples with different total deposition thicknesses (5- and 15-nm) are plotted together in Figure R7b. For the samples with the same thickness, we observe a similar spectra shape and transmission magnitude, which implies that approximately equal amounts of materials have been deposited. Upon the hydrogenation, $\text{Pd}_{80}\text{Au}_{20}$ NP_5^{50} and NP_{15}^{50} samples behave similar to $\text{Pd}_{80}\text{Ag}_{20}$ NP_5^{50} and NP_{15}^{50} , with comparable optical transmission changes ($\Delta T(\lambda)$) and similar spectra shape (Figure R7c). However, the plateau pressures found in $\Delta T(\lambda)$ optical isotherms of PdAu samples are at higher pressure regimes ($P_{\text{Abs}} \approx 7$ mbar) than these of PdAg systems ($P_{\text{Abs}} \approx 1$ mbar) (Figure R7d). We note that the P_{Abs} we found in PdAg NPs are slightly smaller than that of PdAu nanoparticles with a similar composition ($P_{\text{Abs}} \geq 10$ mbar).¹³⁻¹⁵ In addition, a noticeable hysteresis can be seen in $\Delta T(\lambda)$ isotherms of $\text{Pd}_{80}\text{Au}_{20}$ NP_{15}^{50} ($P_{\text{Hys}} \approx 1.1$ mbar) and $\text{Pd}_{80}\text{Au}_{20}$ NP_5^{50} ($P_{\text{Hys}} \approx 0.7$ mbar), which are wider than these of PdAg samples ($P_{\text{Hys}} < 0.1$ mbar). However, the sensor accuracies calculated over the pressure range of 10^1 to 10^6 μbar are still below 5% level (Figure R7e).

At a given pressure $P_{\text{H}_2} > P_{\text{Abs}} \approx 7$ mbar, the response time (t_{90}) of a PdAu sensor is slightly higher than that of a PdAg sensor with the same thickness, which follows the power-law with a similar slope (Figure 4d). Interestingly, we observe a kink at $P_{\text{Abs}} \approx 7$ mbar, below which t_{90} of PdAu still follows the power law but with a noticeable smaller slope. The t_{90} of $\text{Pd}_{80}\text{Au}_{20}$ NP_{15}^{50} and NP_5^{50} samples at 1 mbar are 13 s and 5 s, respectively, which are much shorter than these of PdAg sensors (45 s and 9 s for NP_{15}^{50} and NP_5^{50} samples, respectively). Since the kink is observed in the middle of the plateau, which is in a similar position as that in the pure Pd sensor case (Figure 2f), we hypothesize that the strain-induced energy barrier in the $\text{Pd}_{80}\text{Au}_{20}$ is still significant at $P_{\text{Abs}} \approx 7$ mbar and is the cause of this observation. We note that the differences in t_{90} between the PdAu and PdAg samples with the same t and P_{H_2} might come from several factors, such as differences in morphologies (see Figure S8 in SI), and hydrogen permeability.¹⁶

Change made to the manuscript:

- We have added the data of PdAu samples to Figure 4. The above discussions have been added to the main-text, at Page 10, Line 215.
- Figure S19 and the associated discussion have been revised with the data of PdAu samples.

11. The noise level in the raw data for Pd₈₀Ag₂₀ and Pd₈₀Co₂₀ presented in Figure S16, the noise level for the Co-alloy system appears a lot higher. Why?

Author Reply: We thank the reviewer for pointing this out. Since the old Figures S16a and b had different $\Delta T/T$ scales (y-axis), the noise level of PdCo system looked a lot higher. When we have now compiled these two figures into the same plot (Figure R8), they show a consistent noise level.

Figure R8. $\Delta T/T$ response of (a) Pd₈₀Ag₂₀ ($t = 5$ nm) and (b) Pd₈₀Co₂₀ ($t = 5$ nm) composite NPs with different hydrogen concentration (C_{H_2}), measured in flowing nitrogen (400 ml/min). Shaded areas denote the periods where the sensor is exposed to hydrogen. Up-down arrows in the insets of (a) and (b) correspond to 0.03 %.

Furthermore, the authors define their detection limit as 3sigma but the actual level of 3 sigma is not included in the figure and it is therefore not possible to check whether the made LoD claim is correct. In particular for the Co system I am not sure if a LoD of 2.5 ppm really can be claimed.

Author Reply: In Figures R8 and old Figure S16, we used a hydrogenation-dehydrogenation cycle of 1.5-3.5 minutes, which might not be long enough for the sensor to reach a distinct response upon hydrogenation. Hence, we used a cycle of 3-minutes hydrogenate and 5-minutes dehydrogenation, as presented in our response to Comment #9. The 3σ levels are also included in the revised figures. We believe that these newly added data and corresponding revisions convincingly support our reported LOD values of 2.5 ppm and <10 ppm for the fresh- and 10-months-old Pd₈₀Co₂₀ NP₅⁵⁰ samples, respectively.

12. The authors compared “vacuum-mode and flow-mode sensing” and find similar sensing response in terms of LoD. Why would they not? Also in flow mode only inert gas background is used. It would, however, from an application perspective be very interesting to see how their sensors perform at more realistic conditions like in air, and how that affects the LoD.

Author Reply: We thank the reviewer for the comment. We have revised Section S9. Noise evaluation in SI, by adding the noise evaluation in flow mode with the synthetic gas. The standard deviations of noise distribution σ and LODs for each condition (vacuum mode, flow mode with nitrogen, and flow mode with synthetic gas) have been spelt out explicitly.

We agree with the reviewer that it is crucial for a hydrogen sensor to perform in a realistic condition like in air. First, we examine the LOD of the representative $\text{Pd}_{80}\text{Co}_{20}\text{NP}_5^{50}$ system in synthetic air carrier gas, as presented in Figure R9. Generally, in synthetic air, the sensor exhibits smaller response at a given C_{H_2} in comparison to the response of the sensor in vacuum/inert carrier gas. This is due to the competing processes of the hydrogen oxidation reaction and the adsorption sites are blocked by O.^{10, 15} This rises the LOD of 2.5-ppm in N_2 air carrier to ~ 10 -ppm in synthetic air gas carrier.

Figure R9. (a) $\Delta T/T$ response of $\text{Pd}_{80}\text{Co}_{20}$ ($t = 5$ nm) composite NPs with different hydrogen concentration (C_{H_2}), measured in flowing synthetic air (400 ml/min). Shaded areas denote the periods where the sensor is exposed to hydrogen. (b) $\Delta T/T$ response of $\text{Pd}_{80}\text{Co}_{20}$ ($t = 5$ nm) composite NPs, measured in vacuum/pure H_2 (square), nitrogen carrier gas (circle), and synthetic air carrier gas (diamond). By defining the limit of detection (LOD) as $3\sigma \approx 0.013\%$ (with $\sigma = 0.0042\%$, is the noise of the acquired signal with synthetic gas carrier air, see Section S9 in SI), marked by the blue dashed line, we achieve the LOD to the range of 2.5 ppm and 10 ppm for the $\text{Pd}_{80}\text{Co}_{20}$ ($t = 5$ nm) sensor in N_2 and synthetic air carrier gases, respectively.

In addition, we also consider the influence of other gases (e.g., CO , CO_2 , CH_4) on PdCo sensors with and without PMMA coating (as suggested by Reviewer #3, Comment #6).

Change made to the manuscript:

- We have revised Figure 5 and the associated discussions with the data from the PdCo sensor (5 nm) only. The PdAg sensor data in the old Figure 5 have been moved to the SI as Figure S22.

- Figures S20 and S21 have been added to SI, which show the sensor signals with different C_{H_2} measured in flowing nitrogen and synthetic gas, respectively.

13. In the conclusions section the authors claim a “groundbreaking level of sensitivity” and that their “sensors demonstrate viable path forward to spark-proof optical sensors for hydrogen detection applications that no other method has been proven to date”. These are very strong statements which, in particular for the first one, are simply not correct. I would therefore strongly advise the authors to not oversell their work in this way since it is absolutely not necessary. Their work is good enough as it is.

Author Reply: We thank the reviewer for the insightful suggestion, and have removed this claim in the revised manuscript. The conclusion has been revised accordingly.

Change made to the manuscript: We have removed the “groundbreaking level of sensitivity” claim from the Conclusions part.

Responses to Comments from Reviewer #2

- What are the major claims of the paper?

This paper claims to have created optical hydrogen sensors with a 2.5 ppm limit of detection with a response time of 0.85 second at 1 mbar by utilizing novel nanostructures to increase the surface to volume ratio of the metal hydride. By investigating different deposition thicknesses and angles, as well as different alloying combinations, they were able to optimize their sensor.

- Are the claims novel? If not, please identify the major papers that compromise novelty. This proposed nano-achitecture is novel to my knowledge.

- Will the paper be of interest to others in the field?

Yes.

- Will the paper influence thinking in the field?

Yes, this novel nanostructure could inspire more sensors similar to its design.

Author Reply: We thank the reviewer for the positive assessment of our work. We are happy to have an opportunity to address the reviewer's critical remarks, important questions, and specific suggestions.

Questions:

1. In Figure 2, you should show the data for $t_{pd} = 0$ (bare PS beads) for each subplot if you are going to explicitly refer to it in the paper.

Author Reply: We thank the reviewer for the useful suggestion. The data for $t_{pd} = 0$ nm (bare PS substrate) has been added.

Figure R10. (a) Experimental optical transmission spectra $T(\lambda)$ (at $P_{H_2} = 0$ mbar) of NP with different t_{pd} .

Change made to the manuscript: We have replaced Figure 2a with Figure R10.

2. Why is there a peak in t_{90} ? Literature is cited to show that this peak is not unusual, but is the physical cause known for this peak?

Author Reply: The peak in the t_{90} versus P_{H_2} relation for P_{Abs} has been reported in some Pd-based hydrogen sensor,⁹⁻¹¹ however, the origin of this observation has not been described. We hypothesize that the peak is associated with the abrupt change in hydride volume during the $\alpha - \beta$ phase transition. Such abrupt change or sudden entropy change occurs only if the hydride system thermodynamically overcomes the large strain-induced energy barrier upon hydrogenation, which requires longer time to reach the equilibrium. The strain-induced energy barrier is relatively strong in pure Pd based sensors, which results in the large t_{90} peak in this pressure regime. It is clear in Figure 2f that such barrier is larger with a larger grain size and film thickness,⁵ and interestingly, the barrier becomes much less significant as the structure transforms from the film-like to the nanoparticle island-like for $t_{Pd} \leq 2$ nm.

Change made to the manuscript: We have added the above discussions in Page 7, Line 144.

3. Is there a reason why the authors chose PdAg and PdCo as their test alloys instead of the more common PdAu?

Author Reply: We thanks the reviewer for raising up this question. We used PdAg, instead of PdAu, firstly due to the availability of the materials for electron beam evaporation at the time of the sample fabrication. In addition, since the hydrogen permeability in PdAg is reported to be faster than that in PdAu,¹⁶ we speculated that we would have a faster response time in PdAg. On the other hand, as mentioned in the main-text, we used PdCo alloy due to the improvement of the kinetics of hydrogenation over a PdAg and PdAu alloy, by (1) remaining in the α -phase over the pressure region of interest, which removes the kinetic steps of the α - to β -phase transition and subsequent hydrogen atomic diffusion through the β -phase,¹⁷ and (2) cobalt offers a greater metal-hydrogen bond strength than silver, which could facilitate dissociative chemisorption of hydrogen.¹⁸

What results would they expect from experiments using PdAu with their nanostructure?

Author Reply: The optical properties and sensing performance of a similar sensor architecture using PdAu have been discussed throughout in the Comment #10 of Reviewer #1. We have revised the manuscript and the SI with the new data of PdAu systems.

Change made to the manuscript:

- We have added the data of PdAu samples to Figure 4. The discussions for PdAu sample data have been added to the main-text, at Page 10, Line 215.
- Figure S19 and the associated discussion have been revised with the data of PdAu samples.

Along with this, why did the authors choose 80/20 alloy ration over other options?

Author Reply: The 80/20 alloy ratio was chosen for our NP system, in order to compromise between the sensor accuracy, response time, and LOD. A higher composition of Pd (>80 %) allows a greater signal sensor and better LOD, however, the sensor is not “hysteresis-free” and it would be impossible to reach

$t_{90} \leq 1$ s at 1 mbar.¹³ Conversely, over-reducing composition of Pd to eliminate the hysteresis and to improve response time will sacrifice optical contrast, resulting in a poor LOD.

Recent work on Pd alloys for hydrogenation should be cited, e.g. Palm et al, ACS Applied Materials & Interfaces 11, 45057-45067 (2019) and Bannenberg, et al, ACS Appl. Mater. Interfaces 2019, 11 (17), 15489–15497.

Author Reply: We thank the reviewer for referring us to these useful papers, which have been added as Reference 45 and 47 in the main text of the revised manuscript.

4. How do the authors determine the start time of hydrogen exposure? Is it defined as the time that the gas is switched over, or is there any accounting for gas change over time within their chamber?

Author Reply: In order to determine the start time of hydrogen exposure for the response time measurement, we record the pressure level of the chamber in real time (with associated computer's time stamp), by a digital pressure transducer (~1000 readings/second). This pressure transducer is placed relatively at the same position with the testing sample, and they have approximately equal gas travel distances from the hydrogen control valve. Using **the same computer**, we record in real time the transmission changes of the sample with associated time stamps. In the data analysis, we match the time stamps from the pressure transducer and transmission measurement to determine the start time of hydrogen exposure.

5. Line 50, what limit of detection are you drawing the line at? Nugroho et al. Nat. Mater. 18, 489–495 (2019) claims a $t_{90} = 1$ sec sensing time at 1 mbar exposure with a LOD of 3 ppm in an Ar atmosphere.

Author Reply: We thank the reviewer for this interesting question and also for referring us to the useful reference. We agree that in *Nat. Mater.* 18, 489–495 (2019),¹² Nugroho *et al.* claims a (i) $t_{90} = 1$ sec at 1 mbar and (ii) a LOD of 3 ppm with Ar carrier gas. However, these two impressive features were not achieved within a **single** sensor (as also mentioned in Table S1 in the SI). In particular, (i) the LOD of 3 ppm was observed with 190×25 (nm²), but the response time at 1 mbar for this sensor was reported at > 2 sec. On the other hand, by downsizing the sensor to 100×25 (nm²), (ii) $t_{90} = 1$ sec at 1 mbar was achieved, but the LOD was expected to be at higher concentrations (e.g., the LOD of the 100×25 nm² sensor was not mentioned throughout the manuscript). Therefore, we would like to highlight the capability of achieving both feats (i) and (ii) in the Pd₈₀Co₂₀ NP₅⁵⁰ system, as one of the key findings in our present paper.

Responses to Comments from Reviewer #3

The authors reported on a compact optical hydrogen sensing platform, which was comprised of Pd and Pd-alloy nano-patchy arrays. The hexagonally packed nanoscale arrays were fabricated by single-metal glancing-angle deposition on polystyrene (PS) nanosphere monolayers. This fabrication process was simple and with no post-processing required. The tunable film thicknesses, patchy diameters and shapes (hemispheres or donuts) resulted in tunable and rapid response rates. By alloying with Co, the response time of the metasurfaces was reduced below 0.85 s from 1-100 mbar of H₂ partial pressure, surpassing the strictest requirements for H₂ sensing while preserving good accuracy (<2.5% full scale) and 2.5-ppm limit of detection (LOD).

The demonstrations of the hydrogen sensing performances of Pd and Pd-alloy nanostructure arrays have been reported in many previous works. The detection limits of hydrogen concentrations can be down to 100 ppm (ACS Sens. 2020, 5, 4, 917), and PdAg nanocomposite exhibits a high sensitivity of 5.5% for 1.6% H₂ concentration (ACS Sens. 2020, 5, 2367). The experiments were carefully performed and the results were convincing. However, although the hydrogen sensing performance was enhanced, the measurements were only performed in a lab experiment manner. It is far away from the practical application. I do not think the novelty or the importance of this work is sufficient for Nature Communications. It is more suitable for a journal on nanoscience or materials.

Author Reply: We thank the reviewer for the critical assessment of our work and for the insightful and instructive comments/suggestions. We have studied all the comments of the reviewers carefully and performed several new experiments to address these comments. For instance, we have added the new results (the sensor's long-term stability in ambient air, sensing performance with different temperatures and relative humidity, and deactivation tests with CO, CO₂, and CH₄) to the revised manuscript. We hope that these important additions would justify the reviewer's concern regarding the practical application of our sensor.

1. What are the diameters of the PS nanospheres in Figure 1? Why were the PS nanospheres with such diameters chosen?

Author Reply: The diameter of the PS nanosphere in Figure 1 and throughout the manuscript is $D = 500$ nm, which is described by the scale bars.

In order to achieve the largest surface-to-volume ratio (SVR) for the patchy particles (which is desired for faster absorption/desorption kinetics), a small diameter of nanosphere template is required. In addition, the hexagonally close-pack monolayer template should contain a low level of defect, as these defects can allow the materials to be deposited directly to glass substrate in metal glancing angle deposition (GLAD), which can potentially affect the optical response and slow down response time of the patchy particle.⁹ In addition, the highly-uniform nanosphere monolayer template also aids to improve the reproducibility of the sensor fabrication and sensing performance.

In our fabrication process, the PS nanosphere monolayers were prepared by the air-water interface method,¹⁹⁻²⁴ and the diameter of nanosphere can be varied from 200 nm to several μm . With the two smallest nanosphere sizes ($D = 200$ nm and $D = 350$ nm), we have not been successful to achieve a low-

defect hexagonally close-pack monolayer. Typical scanning electron microscopy (SEM) micrographs of these monolayers can be seen in Figure R11. On the other hand, we have been consistently achieving a very high quality with low defect PS (<0.8 %) monolayer with nanosphere $D = 500$ nm (see SI of Ref. ²¹). Therefore, this diameter of $D = 500$ nm has been chosen. We note that we have been improving the quality of the monolayer template with smaller size ($D = 200$ nm and $D = 350$ nm), which can be served as a fabrication template for our future works.

Figure R11. Scanning electron microscopy (SEM) micrographs of PS nanosphere monolayers with (a) $D = 200$ nm and (b) $D = 350$ nm. Scale bar: $4 \mu\text{m}$.

Change made to the manuscript: We have explicitly described the nanosphere diameter in the caption of Figure 1.

2. What are the morphologies of the pure Pd NP samples simulated in Figure 3a? What does the scale bar stand for?

Author Reply: We thank the referee for bringing this issue for further discussions. The morphologies of the NP samples simulated in Figure 3a are thickness distribution the hemisphere cap, based on a simple simulation including a uniform vapor flux approaches in direction $\hat{l}(\theta_0, \varphi_0)$ to an array of hexagonal close-packed nanosphere with diameter $D = 500$ nm (Figure R12). We apologize for not providing the explanation how the simulations were performed in the previous version of the manuscript. Thus, a detailed explanation has been added to the SI (Section 1), which aids the readers to assess the results easily.

Figure R12. (a) A cartoon illustrates the simulated vapor deposition on an array of nanospheres.

The scale bars in SEM images in Figure 3a correspond to 500 nm. The color bar on the right side of Figure 3a represents the deposited thickness (normalized for the maximum thickness), based on the simulation we mentioned above.

3. The top-view SEM images and corresponding simulated morphologies of the pure Pd NP samples fabricated with a vapor incident angle of $\theta = 50$ degrees should be provided in Figure 3a.

Author Reply: We thank the reviewer for the useful suggestion. The top-view SEM images and corresponding simulated morphologies of the Pd NP₅⁵⁰ sample have been added in Figure 3a.

Change made to the manuscript: The top-view SEM images and corresponding simulated morphologies of the Pd NP₅⁵⁰ sample have been added in Figure 3a. In addition, we have clearly described the meaning of the scale bar and color bar in the caption.

4. The optical hydrogen absorption spectrum, sensor accuracy and response time of the Pd₈₀Co₂₀ ($t = 15$ nm) should be provided in Figure 4 for comparison.

Author Reply: We thank the reviewer for this comment. The optical hydrogen absorption spectrum, isotherm, sensor accuracy and response time of Pd₈₀Co₂₀ NP₁₅⁵⁰ have been added. In the revised manuscript, we also added these data of Pd₈₀Au₂₀ NP₁₅⁵⁰ and NP₅⁵⁰ samples, following the suggestions of Reviewer #1 (Comment #10) and Reviewer #2 (Comment #3).

Change made to the manuscript: The optical isotherm, sensor accuracy and response time of Pd₈₀Co₂₀ NP₁₅⁵⁰ have been added to Figure 4. The optical hydrogen absorption spectrum has been added to Figure S19.

5. The authors are suggested to give a standard curve of the optical response as a function of the hydrogen concentration.

Author Reply: We thank the referee for the suggestion. The optical transmission spectra versus hydrogen pressure figures have been added in the SI.

Change made to the manuscript: The spectra responses of Pd and Pd-composite hydrogen sensor to an increasing/decreasing hydrogen pressure have been added to SI as Figures S23-S25.

6. The authors investigated the optical response of the Pd-alloy nano-patchy arrays in vacuum and nitrogen. They did not consider the disturbances to the hydrogen sensing in practical situations, such as the influences from temperature, moisture, and other gases (e.g., CO, CO₂, air, etc.).

Author Reply: We agree with the reviewer and therefore have carried out the deactivation tests in synthetic air carrier gas to mimic real application conditions. An excellent selectivity of Pd₈₀Co₂₀ NP₅⁵⁰ to high-concentration pulses of CO₂ (5%) and CH₄ (5%) can be observed in Figures R13a and b. However,

upon exposure to high-concentration pulses of CO (0.2%), the sensor signal dropped considerably (~40% magnitude, but not completely deactivated), along with the increases of response times and release times. In addition, the Pd₈₀Co₂₀ NP₅⁵⁰ sensor shows a good tolerance with temperature up to 315 K (Figures R13c and e) and medium resistances to humidity (Figures R13d and f).

Figure R13. (a) Time-resolved $\Delta T/T$ response of Pd₈₀Co₂₀ NP₅⁵⁰ to 2 pulses of 2% H₂ followed by 9 pulses of 2% H₂ + 5% CH₄, 2% H₂ + 5% CO₂, and 2% H₂ + 0.2% CO; and (b) normalized sensor signal to the one obtained with 2% H₂ in synthetic gas flow. The error bars denote the standard deviation from 9 cycles. (c) Time-resolved $\Delta T/T$ response of Pd₈₀Co₂₀ NP₅⁵⁰ to 9 pulses of 2% H₂ with different temperature and (e) normalized sensor signal one obtained with 2% H₂ in synthetic gas flow at 296 K. (d) Time-resolved $\Delta T/T$ response of Pd₈₀Co₂₀ NP₅⁵⁰ to 9 pulses of 2% H₂ with different relative humidity (RH) and (f) normalized sensor signal one obtained with 2% H₂ in dry condition. All measurements were performed using synthetic gas as carrier gas.

The results of the deactivation tests provide convincing evidence for the slow-degradation of sensing performances of Pd₈₀Co₂₀ NP₅⁵⁰ over times (t_{90} at 1 mbar and LOD are 0.85/2.5 s and 2.5/<10 ppm for a fresh/10-months-old samples, respectively, see Comment #9 of Reviewer #1). Therefore, we seek a polymer-coating layer which protects the sensing layer against humidity, CO, and other toxic gases. Inspired by previous works,^{12, 25-26} we coated the Pd₈₀Co₂₀ NP₅⁵⁰ sample with a ~50-nm layer of polymethyl methacrylate (PMMA) (by spin-coating PMMA dissolved in acetone, more details can be found in Methods Section in the main-text). Similar deactivation tests were carried out on this Pd₈₀Co₂₀

NP₅⁵⁰ sample, and the results are summarized in Figure R14. We note that coating PMMA does not vitiate the sensing capabilities of this sensor (see Figure S30 in SI). As we can see in Figure R14, the PMMA-coated layer provides an excellent protection of the sensor against CO and humidity. This enhances our observations in long-term sensing performance tests (see Comment #9 of Reviewer #1), as this PMMA-coated layer can sufficiently mitigate the degradation of sensing performance when this sensor operates in a practical condition. We hope that the newly-added results of the Pd₈₀Co₂₀ NP₅⁵⁰ and Pd₈₀Co₂₀ NP₅⁵⁰/PMMA sensors (including the sensors' long-term stability in ambient air, sensing performance with different temperature and relative humidity, and deactivation tests with CO, CO₂, and CH₄) would justify the practical application of our sensors.

Figure R14. (a) Time-resolved $\Delta T/T$ response of Pd₈₀Co₂₀ NP₅⁵⁰/PMMA to 2 pulses of 2% H₂ followed by 9 pulses of 2% H₂ + 5% CH₄, 2% H₂ + 5% CO₂, and 2% H₂ + 0.2% CO; and (b) normalized sensor signal to the one obtained with 2% H₂ in synthetic gas flow. The error bars denote the standard deviation from 9 cycles. (c) Time-resolved $\Delta T/T$ response of Pd₈₀Co₂₀ NP₅⁵⁰/PMMA to 9 pulses of 2% H₂ with different relative humidity (RH) and (d) normalized sensor signal one obtained with 2% H₂ in dry condition. All measurements were performed using synthetic gas as carrier gas.

Change made to the manuscript:

- Figures R13 and R14 have been added in the main-text as Figure 6 and 7, respectively. A paragraph has been added in Page 15, Line 319 to describe these figures.
- The Abstract and Conclusions have been revised accordingly.

7. The error bar should be included in Figure 5c.

Author Reply: We thank the reviewer for the useful suggestion. The error bar has been included in Figure S22 (the old Figure 5c) and the new Figure 5c.

Change made to the manuscript: The error bar has been included in Figure S22 (the old Figure 5c) and the new Figure 5c.

Reference

1. Zhao, G.; Wang, W.; Bae, T.-S.; Lee, S.-G.; Mun, C.; Lee, S.; Yu, H.; Lee, G.-H.; Song, M.; Yun, J., Stable ultrathin partially oxidized copper film electrode for highly efficient flexible solar cells. *Nat. Commun.* **2015**, *6* (1), 1-8.
2. Lazzari, R.; Jupille, J., Silver layers on oxide surfaces: morphology and optical properties. *Surface science* **2001**, *482*, 823-828.
3. Larsen, G. K.; He, Y.; Wang, J.; Zhao, Y., Scalable fabrication of composite Ti/Ag plasmonic helices: controlling morphology and optical activity by tailoring material properties. *Adv. Opt. Mater.* **2014**, *2* (3), 245-249.
4. Sykes, M. F.; Essam, J. W., Exact critical percolation probabilities for site and bond problems in two dimensions. *Journal of Mathematical Physics* **1964**, *5* (8), 1117-1127.
5. Berube, V.; Radtke, G.; Dresselhaus, M.; Chen, G., Size effects on the hydrogen storage properties of nanostructured metal hydrides: A review. *International Journal of Energy Research* **2007**, *31* (6□7), 637-663.
6. Jimenez, G.; Dillon, E.; Dahlmeyer, J.; Garrison, T.; Garrison, T.; Darkey, S.; Wald, K.; Kubik, J.; Paciulli, D.; Talukder, M., A comparative assessment of hydrogen embrittlement: palladium and palladium-silver (25 weight% silver) subjected to hydrogen absorption/desorption cycling. *Advances in Chemical Engineering and Science* **2016**, *6* (03), 246.
7. Yin, S.; Cheng, G.; Chang, T.-H.; Richter, G.; Zhu, Y.; Gao, H., Hydrogen embrittlement in metallic nanowires. *Nat. Commun.* **2019**, *10* (1), 1-9.
8. Xie, D.; Li, S.; Li, M.; Wang, Z.; Gumbsch, P.; Sun, J.; Ma, E.; Li, J.; Shan, Z., Hydrogenated vacancies lock dislocations in aluminium. *Nat. Commun.* **2016**, *7* (1), 1-7.
9. Luong, H. M.; Pham, M. T.; Madhogaria, R. P.; Phan, M.-H.; Larsen, G. K.; Nguyen, T. D., Bilayer Plasmonic Nano-lattices for Tunable Hydrogen Sensing Platform. *Nano Energy* **2020**, *71*, 104558.
10. Yang, F.; Kung, S.-C.; Cheng, M.; Hemminger, J. C.; Penner, R. M., Smaller is faster and more sensitive: the effect of wire size on the detection of hydrogen by single palladium nanowires. *ACS Nano* **2010**, *4* (9), 5233-5244.
11. Herkert, E.; Sterl, F.; Strohfeltdt, N.; Walter, R.; Giessen, H., Low-Cost Hydrogen Sensor in the ppm Range with Purely Optical Readout. *ACS sensors* **2020**, *5* (4), 978-983.
12. Nugroho, F. A.; Darmadi, I.; Cusinato, L.; Susarrey-Arce, A.; Schreuders, H.; Bannenberg, L. J.; da Silva Fanta, A. B.; Kadkhodazadeh, S.; Wagner, J. B.; Antosiewicz, T. J., Metal-polymer hybrid nanomaterials for plasmonic ultrafast hydrogen detection. *Nature materials* **2019**, *18* (5), 489.
13. Wadell, C.; Nugroho, F. A. A.; Lidström, E.; Iandolo, B.; Wagner, J. B.; Langhammer, C., Hysteresis-free nanoplasmonic Pd-Au alloy hydrogen sensors. *Nano Lett.* **2015**, *15* (5), 3563-3570.
14. Westerwaal, R.; Rooijmans, J.; Leclercq, L.; Gheorghe, D.; Radeva, T.; Mooij, L.; Mak, T.; Polak, L.; Slaman, M.; Dam, B., Nanostructured Pd-Au based fiber optic sensors for probing hydrogen concentrations in gas mixtures. *International journal of hydrogen energy* **2013**, *38* (10), 4201-4212.
15. Darmadi, I.; Nugroho, F. A. A.; Kadkhodazadeh, S.; Wagner, J. B.; Langhammer, C., Rationally-Designed PdAuCu Ternary Alloy Nanoparticles for Intrinsically Deactivation-Resistant Ultrafast Plasmonic Hydrogen Sensing. *ACS sensors* **2019**, *4* (5), 1424-1432.
16. Knapton, A., Palladium alloys for hydrogen diffusion membranes. *Platinum Metals Review* **1977**, *21* (2), 44-50.
17. Nahm, K.; Kim, W.; Hong, S.; Lee, W., The reaction kinetics of hydrogen storage in LaNi5. *International Journal of Hydrogen Energy* **1992**, *17* (5), 333-338.
18. Christmann, K., Some general aspects of hydrogen chemisorption on metal surfaces. *Progress in surface science* **1995**, *48* (1-4), 15-26.
19. Larson, S. R.; Luong, H.; Song, C.; Zhao, Y., Dipole Radiation Induced Extraordinary Optical Transmission for Silver Nanorods Covered Silver Nanohole Arrays. *J. Phys. Chem. C* **2019**, *123* (9), 5634-5641.

20. Larson, S.; Carlson, D.; Ai, B.; Zhao, Y., The Extraordinary Optical Transmission and Sensing Properties of Ag/Ti Composite Nanohole Arrays. *Physical Chemistry Chemical Physics* **2019**.
21. Luong, H. M.; Pham, M. T.; Ai, B.; Nguyen, T. D.; Zhao, Y., Magnetoplasmonic properties of Ag-Co composite nanohole arrays. *Phys. Rev. B* **2019**, *99* (22), 224413.
22. Luong, H. M.; Minh, P. T.; Nguyen, T.; Zhao, Y., Magneto-plasmonic properties of Ag-Co composite nano-triangle arrays. *Nanotechnology* **2019**, *30* (42), 425203.
23. Luong, H. M.; Pham, M. T.; Larsen, G. K.; Nguyen, T. D., Plasmonic sensing of hydrogen in Pd nano-hole arrays. *Plasmonics: Design, Materials, Fabrication, Characterization, and Applications XVII* **2019**, 11082, 110821D.
24. Luong, H. M.; Pham, M. T.; Nguyen, T. D.; Zhao, Y., Enhanced Resonant Faraday Rotation in Multilayer Magnetoplasmonic Nanohole Arrays and Their Sensing Application. *J. Phys. Chem. C* **2019**, *123* (46), 28377-28384.
25. Hong, J.; Lee, S.; Seo, J.; Pyo, S.; Kim, J.; Lee, T., A highly sensitive hydrogen sensor with gas selectivity using a PMMA membrane-coated Pd nanoparticle/single-layer graphene hybrid. *ACS Appl. Mater. Interfaces* **2015**, *7* (6), 3554-3561.
26. Min, K.; Paul, D., Effect of tacticity on permeation properties of poly (methyl methacrylate). *Journal of Polymer Science Part B: Polymer Physics* **1988**, *26* (5), 1021-1033.

Reviewers' Comments:

Reviewer #1:

Remarks to the Author:

The authors have done a very thorough work in addressing my concerns (and in my opinion also the two other reviewer's comments) and I can now wholeheartedly recommend publication in Nature Communications.

Reviewer #2:

Remarks to the Author:

The authors have adequately addressed my concerns.

Reviewer #3:

Remarks to the Author:

The authors have addressed all of the questions raised by the three reviewers. I found that their answers are scientifically reasonable. The authors have also performed new experiments to support their statements and strengthen their work. I think their answers are satisfying. The quality of the manuscript has been improved a lot.

Originally I did not think this work is new or important enough. However, since the other two reviewers are positive about the work and the authors have carefully addressed all of the comments, I would change my mind. I think this work is publishable now on Nature Communications.

Reviewer #1

The authors have done a very thorough work in addressing my concerns (and in my opinion also the two other reviewer's comments) and I can now wholeheartedly recommend publication in Nature Communications.

Reviewer #2

The authors have adequately addressed my concerns.

Reviewer #3

The authors have addressed all of the questions raised by the three reviewers. I found that their answers are scientifically reasonable. The authors have also performed new experiments to support their statements and strengthen their work. I think their answers are satisfying. The quality of the manuscript has been improved a lot.

Originally I did not think this work is new or important enough. However, since the other two reviewers are positive about the work and the authors have carefully addressed all of the comments, I would change my mind. I think this work is publishable now on Nature Communications.

Author Reply: We thank the reviewers for the time and effort that they invested into the review of our manuscript, and for suggesting our work for publication.